# Synergistic Modulation of Seed Metabolites and Enzymatic Antioxidants Tweaks Moisture Stress Tolerance in Non-Cultivated Traditional Rice Genotypes during Germination

**DOI:** 10.3390/plants11060775

**Published:** 2022-03-14

**Authors:** Asish Kanakaraj Binodh, Sugitha Thankappan, Anupriya Ravichandran, Debasis Mitra, Senthil Alagarsamy, Periyasamy Panneerselvam, Ansuman Senapati, Rokayya Sami, Amina A. M. Al-Mushhin, Amani H. Aljahani, Amal Alyamani, Mohammed Alqurashi

**Affiliations:** 1Centre for Plant Breeding and Genetics, Tamil Nadu Agricultural University, Coimbatore 641003, India; 2School of Agriculture and Biosciences, Karunya Institute of Technology and Sciences, Karunya Nagar, Coimbatore 641114, India; sugithat@gmail.com; 3Department of Plant Breeding and Genetics, Agricultural College & Research Institute, Tamil Nadu Agricultural University, Killikulam 628252, India; 22anuabi@gmail.com; 4Crop Production Division, ICAR-National Rice Research Institute, Cuttack 753006, India; debasismitra3@gmail.com (D.M.); asenapati89@gmail.com (A.S.); 5Department of Crop Physiology, Tamil Nadu Agricultural University, Coimbatore 641003, India; senthil.a@tnau.ac.in; 6Department of Food Science and Nutrition, College of Sciences, Taif University, P.O. Box 11099, Taif 21944, Saudi Arabia; 7Department of Biology, College of Science and Humanities in Al-Kharj, Prince Sattam Bin Abdulaziz University, Al-Kharj 11942, Saudi Arabia; a.almushhin@psau.edu.sa; 8Department of Physical Sport Science, College of Education, Princess Nourah bint Abdulrahman University, P.O. Box 84428, Riyadh 11671, Saudi Arabia; ahaljahani@pnu.edu.sa; 9Department of Biotechnology, Faculty of Sciences, Taif University, P.O. Box 11099, Taif 21944, Saudi Arabia; a.yamani@tu.edu.sa (A.A.); m.khader@tu.edu.sa (M.A.)

**Keywords:** hydrolytic enzymes, landraces, moisture-stress tolerance, PEG, phytohormones, ROS-signaling

## Abstract

Traditional rice landraces are treasures for novel genes to develop climate-resilient cultivars. Seed viability and germination determine rice productivity under moisture stress. The present study evaluated 100 rice genotypes, including 85 traditional landraces and 15 improved cultivars from various agro-ecological zones of Tamil Nadu, along with moisture-stress-susceptible (IR 64) and moisture-stress-tolerant (IR 64 Drt1) checks. The landraces were screened over a range of osmotic potentials, namely (−) 1.0 MPa, (−) 1.25 MPa and (−) 1.5 MPa, for a period of 5 days in PEG-induced moisture stress. Physio-morphological traits, such as rate of germination, root and shoot length, vigor index, R/S ratio and relative water content (RWC), were assessed during early moisture stress at the maximum OP of (−) 1.5 MPa. The seed macromolecules, phytohormones (giberellic acid, auxin (IAA), cytokinin and abscisic acid), osmolytes and enzymatic antioxidants (catalase and superoxide dismutase) varied significantly between moisture stress and control treatments. The genotype Kuliyadichan registered more IAA and giberellic acid (44% and 35%, respectively, over moisture-stress-tolerant check (IR 64 Drt1), whereas all the landraces showed an elevated catalase activity, thus indicating that the tolerant landraces effectively eliminate oxidative damages. High-performance liquid chromatography analysis showed a reduction in cytokinin and an increase in ABA level under induced moisture stress. Hence, the inherent moisture-stress tolerance of six traditional landraces, such as Kuliyadichan, Rajalakshmi, Sahbhagi Dhan, Nootripathu, Chandaikar and Mallikar, was associated with metabolic responses, such as activation of hydrolytic enzymes, hormonal crosstalk, ROS signaling and antioxidant enzymes (especially catalase), when compared to the susceptible check, IR 64. Hence, these traditional rice landraces can serve as potential donors for introgression or pyramiding moisture-stress-tolerance traits toward developing climate-resilient rice cultivars.

## 1. Introduction

Global climatic-change impacts, such as prolonged moisture stress, water scarcity and changing precipitation pattern, are the major challenges to agriculture. By 2050, the global demand for agricultural production is expected to increase by 70 percent due to the rising population [1]. Rice (*Oryza sativa* L.) is the key staple in the daily food basket of more than 70 percent of the Indian population. Nevertheless, rice is grown in a wide range of ecosystems, including flood-prone and moisture-stress-prone environments; even a small reduction in rice production may pose a considerable threat to food security [2]. In Asian countries, moisture stress is one of the predominant constraints adversely affecting rice productivity, as it prevails over varied lengths of time and intensity, irrespective of crop growth and development stages [3]. Severe moisture stress during the reproductive and grain-filling phase reduces the economic yield to 48–94% and 60%, respectively, in rice [4]. Climate changes bringing forth frequent and severe moisture-stress episodes emphasize the need to understand the root phenome in response to moisture stress [5,6].

The physiological and biochemical responses of seed during germination at moisture stress impact the survival rate and vegetative growth of the seedlings, and this, in turn, affects the yield and quality. Water deficit drastically reduces the seed germination rate and causes delay in breaking up dormancy, as well as late initiation of the germination process. Biochemical constituents of seeds, such as starch (70–80%), protein (15%), lipids (5%) and free amino acid content, are key molecules in regulating plant responses to abiotic stresses (drought and salinity) [7]. Remobilization of starch and protein during moisture stress supports maintaining the turgidity of cells. The released sugars and other derived metabolites support plant growth under stress and maintain redox-homeostasis by acting as osmoprotectants [8,9] Seed imbibition triggers the synthesis of various hydrolytic enzymes in the aleurone layer or scutellum in response to germination signals. Hydrolytic enzymes convert the storage reserve of seeds into simple forms to facilitate uptake by the embryo. The elevated oxygen levels during hydrolysis activate mitochondrial enzymes involved in Kreb’s cycle (provide precursors for various metabolic reactions) and electron transport chain [10,11]. Amylase synthesized by de novo biosynthetic pathways stimulates the stored starch mobilization and is high until the young plant initiates photosynthesis [12]. Hydrolysis of stored protein in seeds involves peptidases, such as cysteine, serine, aspartic, and metalloproteases, wherein free amino acids are produced by activating protein synthesis in endosperm and embryo [13]. Among the proteases, cysteine proteases in the scutellum, aleurone and endosperm are involved in the mobilization of seed proteins during germination in cereals, specifically in rice [14]. Cysteine proteases participates in protein maturation and elimination of unnecessary synthesized endogenous proteins [15,16]. Lipases are concerned with lipid metabolism that catalyzes β-oxidation, releasing short-chain fatty acids and organic alcohols [17].

Endogenous phytohormones are related to the acclimatization of plants against environmental adversities by mediating source-sink transitions and nutrient allocation [18,19]. Phytohormones, such as auxin (IAA), abscisic acid (ABA), cytokinins (CKs), ethylene (ET), gibberellins (GAs), salicylic acid (SA), brassinosteroids (BRs), jasmonates (JAs) and strigolactones (SL), control a variety of cellular processes and coordinate signal transduction pathways in response to abiotic stress [20]. GA/ABA crosstalk is a key mechanism to cope with early moisture stress. ABA interferes with cell-wall loosening and inhibits water uptake, whereas GAs represses ABA effect by stimulating cell-wall loosening enzymes, such as α-expansins, in the early germination phase [21]. Thus, a rapid decrease in endogenous ABA during seed germination under moisture stress is one of the factors influencing the seed germination rate [22].

Several biochemical and cellular processes associated with germination, such as metabolic reactivation, cellular respiration and mitochondrial biogenesis, translation and/or degradation of stored mRNAs, DNA repair, transcription and translation of new mRNAs, and reserve mobilization, are triggered during seed imbibition [23]. Accumulation of reactive oxygen species (ROS), such as hydrogen peroxide (H_2_O_2_) as a result of water stress, acts as cellular messengers or signaling cues or toxic moieties causing seed vigor losses [24]. However, activation of antioxidant systems during the late phase of germination maintains ROS homeostasis. Interaction of ROS with ABA and GA transduction pathways regulates numerous transcription factors in response to stress [25,26]. Compared with the moisture-stress-mitigating strategies of a mature rice plant, moisture-stress-tolerance mechanisms during the germinating phase are poorly interpreted due to various inherent factors, such as genotypes and environment. Although several signaling components regulating germination and abiotic stresses have been identified, the insights on many of the events during seed germination remain to be clarified. The traditional landraces that are not cultivated possess the inherent potential of withstanding long spells of moisture stress, submergence tolerance and resistance to biotic stresses, besides their nutraceutical properties. Germination of seeds in cultivated genotypes during early moisture stress is a major challenge, whereas landraces mitigate the moisture stress through certain physiological and metabolic modulations. We hypothesized that metabolic or physiological adaptions in non-cultivated traditional rice genotypes, accelerate their tolerance/resilience to cope up with early moisture stress. Such moisture-stress-tolerant rice genotypes are a key asset for developing moisture-stress-tolerant high-yielding rice varieties. The study envisaged the morpho-physiological and metabolic responses of traditional rice landraces from Southern Tamil Nadu during seed germination in response to induced moisture stress.

## 2. Materials and Methods

### 2.1. Plant Materials and Experimental Design

The present investigation was undertaken to study the physiological and metabolic factors contributing to the resilience/resistance and susceptibility in traditional rice genotypes against moisture stress. Approximately 100 rice genotypes, including 85 traditional landraces and 15 improved cultivars, from different agro-climatic zones of Tamil Nadu, India (Appendix A), were subjected to in vitro screening for moisture-stress tolerance, along with respective checks (IR 64 Drt1 and IR 64 as moisture-stress tolerant and susceptible controls, respectively). The experiment was conducted at the Department of Plant Breeding and Genetics, Agricultural College and Research Institute, Killikulam, Vallanadu, Tuticorin Dt, Tamil Nadu (8°46′ N latitude and 77°42′ E longitude and 40 m above MSL), during the Rabi season (October–March) of 2019/2020.

Pre-soaked seeds were surface sterilized with sodium hypochlorite (1.0%) for 3 min and thoroughly rinsed 3 to 5 times in distilled water before introducing seeds on the top of moist absorbent paper placed inside the sterilized Petri plates. The seeds were then moistened with sterile water (control—C) or polyethylene glycol (PEG 6000) equivalent to osmotic potentials of (−)1.0, (−)1.25 and (−)1.5 MPa [27]. The experiment was conducted in factorial completely randomized design (CRD), with three replications. Fifty seeds were maintained for each treatment. The sealed Petri plates were incubated for 5 days in a plant-growth chamber at 28 °C, 60% relative humidity and 12 h light intensity (200 moles m^−2^ s^−1^). Seeds were considered to be germinated at the maximum stress intensity, when the radicle length was at least 2 mm long.

### 2.2. Germination Rate (GR) and Vigor index (VI)

The average number of seeds germinated on 5 days after incubation (DAI) was accounted for calculating the germination rate and vigor index [28].
$$Germination\;rate=\frac{Number\;of\;Germinated\;Seeds}{Number\;of\;Germination\;Days}$$
$$Vigour\;Index=\left(Average\;Shoot\;Length+Average\;Root\;length\right)\;\times Germination\;Percentage$$

### 2.3. Relative Water Content (RWC)

Fresh weight, turgid weight and dry weight of the seedlings measured under C and S conditions were used to calculate the relative water content [29] according to the following equation:$$Relative\;Water\;Content=\frac{Fresh\;Weight-Dry\;Weight}{Turgid\;Weight-Dry\;Weight}\times 100$$

### 2.4. Seed Carbohydrates and Protein

#### 2.4.1. Starch

Starch content in the germinated seeds was determined according to Clegg [30]. Fresh samples (0.5 g per sample) were homogenized in hot 80% ethanol to remove sugars and dried in a water bath. Extraction buffer contained 5.0 mL water and 6.5 mL 52% fresh perchloric acid. The homogenate was centrifuged, and the supernatant was pooled after repeating the extraction. The assay mixture contained 0.2 mL of crude extract, 0.8 mL distilled water and 4 mL anthrone reagent. The intensity of green to dark green color was measured colorimetrically at 630 nm against water as blank and expressed in g 100 g^−1^ FW. Total soluble sugar content in the seed samples was determined according to the method of Yemn and Willis [31].

#### 2.4.2. Protein

Fresh samples (500 mg) were extracted in ice-cold phosphate buffer (5–10 mL). The crude extract was centrifuged at 10,000 rpm for 10 min at 4 °C, and the supernatant was maintained at 0–4 °C. To 0.2 mL of the extract, alkaline copper solution (5 mL) and 0.5 mL of freshly prepared Folin’s reagent were added, followed by incubation in the dark for 30 min at room temperature. Absorbance was recorded at 660 nm by spectrophotometer against the blank. A standard curve representing 40–200 μg mL^−1^ of bovine serum albumin (BSA) was used to calculate the protein content and expressed in mg g^−1^ FW [32].

#### 2.4.3. Total Free Amino Acids

The total free amino acid was estimated by the Ninhydrin method [33]. Exactly 500 mg of the sample was ground with a small quantity of acid-washed sand, followed by extraction in 10 mL of 80% ethanol. The supernatant extracted was used for the quantitative estimation of total free amino acids. To 1 mL of the extract, equal volume of ninhydrin was added, and the volume made up to 2 mL with distilled water. Tubes were kept in a boiling water bath for 20 min; then 5 mL of diluent (equal volume of water and n-propanol) was added, and it was incubated at room temperature for 15 min. The absorbance was read in a colorimeter at 570 nm against a reagent blank made with 0.1 mL of 80% ethanol. A standard curve representing 10–100 μg mL^−1^ of leucine was used to calculate the total free amino acid content and expressed in μg g^−1^ FW.

### 2.5. Quantitative Assay of Hydrolytic Enzymes

#### 2.5.1. Alpha-Amylase

The α-amylase activity in the germinated seeds (5 DAI) was determined according to Muscolo et al. [34]. Exactly 0.5 g of germinated seeds was homogenized in 1:4 *w*/*v* distilled water, and the extract was centrifuged at 14,000 rpm for 30 min at 4 °C. The supernatant was filtered through muslin cloth and then used for the quantitative assay of α-amylase. One unit of amylase activity represents the number of μmoles of reducing sugars formed min^−1^ g^−1^ FW.

#### 2.5.2. Protease

Protease activity was determined according to Harvey and Oaks [35]. The homogenized samples were extracted by using ice-cold acetone mixed with 10 mM Tris-HCl buffer at pH 8.0 and 2M NaCl. The reaction mixture contained 1 mL of crude enzyme extract, 3 mL phosphate buffer and 0.5% casein as substrate. In the control tube, 1 mL of distilled water was added, and the tubes were incubated at 30 °C for 1 h. A standard graph using tyrosinase (0–100 µg) was used to calculate protease activity and absorbance at 660 nm.

#### 2.5.3. Lipase

The homogenized seeds were extracted with twice the volume of ice-cold acetone, followed by an acetone:ether (1:1) wash. The air-dried acetone powder was again extracted in ice-cold water, and the supernatant served as an enzyme source. The samples were titrated against 0.025 N NaOH, using 1% phenolphthalein as an indicator. The volume of titration used for calculating lipase activity was taken from Malik et al. [36]:$$Activity\;meq\;per\;min\;per\;g\;of\;sample=\frac{Volume\;of\;alkali\;consumed\times strength\;of\;alkali}{Weight\;of\;sample\;in\;g\times Time\;\left(min\right)}$$

### 2.6. Determination of Endogenous Phytohormones

#### 2.6.1. Indole Acetic Acid (IAA)

Endogenous IAA of germinating seed on 5 DAI was determined according to Andreae and Van Ysselstein [37]. Fresh sample (500 mg) was homogenized with 5 mL of 95% ethanol, and the supernatant was collected. The precipitate was washed with ethanol, centrifuged and the supernatant collected was pooled. One drop of strong ammonium acetate was added, and the solution was evaporated. After evaporation, 1 mL of 0.1 N NaHCO_3_ solution and 4 mL of Salkowski reagent (2% of 0.5 M FeCl_3_ in 35% perchloric acid) were added, followed by incubation in the dark for 30 min. The absorbance was measured at 530 nm, using a spectrophotometer (SpectraMax^®^ i3x, Sunnyvale, CA, USA). The IAA content was calculated against a standard curve that was prepared by using Indole Acetic Acid and was expressed as µmol g^−1^ FW.

#### 2.6.2. Giberellic Acid (GA)

GA extraction was carried out based on the protocol described by Almeida Trapp et al. [38] and Ghosh et al. [39]. Fresh germinating seed samples were ground to a fine powder in liquid nitrogen. The homogenized mixture in 1 mL of 80% methanol was kept overnight at 4°C, under shaking, and centrifuged at 4 °C, 10,000 rpm, for 15 min. The supernatants collected were dried in a vacuum concentrator. The residues suspended in 500 µL of 80% methanol were used for further analysis. Then 2.0 mL of zinc acetate solution (21.9 g zinc acetate in 80 mL distilled water and 1 mL glacial acetic acid; and volume made up to 100 mL) was added to the dissolved residue, followed by 2.0 mL of potassium ferrocyanide solution (10.6 g in 100 mL water), and the mixture was centrifuged at 10,000 rpm for 10 min. Then 5 mL of supernatant was added with 5 mL of 30% HCl and incubated at 20 °C for 75 min. The absorbance was measured spectrophotometrically at 254 nm.

#### 2.6.3. Quantification of IAA, GA, ABA and Cytokinin, Using HPLC

The samples extracted from germinating seeds subjected to moisture stress and control were quantified by using HPLC (Spectra UV1000, Thermo Fisher, Waltham, MA, USA). The standard stock solutions (HiMedia) were prepared by using methanol (500 μg mL^−1^). The standards and samples (20 μL) were injected through a rheodyne injection valve to the C18 column (Merck), and the compounds separated were detected by using a PDA detector. The endogenous IAA, GA, ABA and Cytokinin (CK) contents were determined by comparing the retention time and peak area of the sample with the standards and expressed as μg g^−1^. Exactly 5 μL of sample was injected into HPLC and eluted out by using a gradient of 60 percent acetonitrile (HPLC grade) for 20 min, at the flow rate of 1 mL.min^−1^, and the compounds were detected at 265 nm [40].

### 2.7. Estimation of Seed Proline Content

Seed proline content on 5 DAI was detected based on the method described by Bates et al. [41]. Approximately 300 mg of germinating seed samples homogenized in 5 mL of 3% sulfosalicylic acid was centrifuged at 3000× *g* for 20 min. The supernatant was mixed with 2 mL of glacial acetic acid and 2 mL of ninhydrin. The mixture was boiled at 100 °C for 25 min, followed by adding 4 mL of toluene. Absorbance of the extract was read at 535 nm in a UV–VIS spectrophotometer.

### 2.8. Determination of Antioxidant Enzymes

For superoxide dismutase and catalase (SOD and CAT) assays, a seed sample of exactly 500 mg was homogenized by using 5 mL of ice-cold buffer (50 mM potassium phosphate buffer (pH 7.0), 1 mm EDTA and 1% (*w*/*v*) polyvinyl pyrrolidone) in an ice-cold pestle and mortar. The supernatant collected after centrifuging the homogenate (10,000 rpm for 30 min at 4 °C) was used for enzyme assays. The reduction in absorbance of H_2_O_2_ in the assay mixture (50 mM phosphate buffer (pH 7.0), 20 mM H_2_O_2_ and 0.1 mL enzyme extract) was observed for 1 min at 240 nm on a microplate reader (Spectramax^®^ i3X, Sunnyvale, CA, USA) and expressed as Units g^−1^ FW min^−1^ [42]. The assay mixture for SOD activity comprised 100 μL of the enzyme, 3 mL of sodium phosphate buffer (50 mM; pH 7.8), 13 mM methionine, 75 μM NBT (nitroblue tetrazolium), 2 μM riboflavin and 0.1 mM EDTA. After incubating the reaction mixture for 15–30 min at 28 ± 2 °C, the absorbance was measured at 560 nm, and activity was expressed as U.g^−1^ FW min^−1^, as described by Beauchamp and Fridovich [43].

### 2.9. Statistical Analysis

All data obtained through the experiment were subjected to the analysis of variance and Duncan’s Multiple Range Test (DMRT) at *p* < 0.05 significance level to evaluate the significant difference in the traits among the genotypes, as performed in R platform (R studio), using the package Agricolae.

## 3. Results

### 3.1. Physiological Changes under Moisture Stress

The present investigation dealt with 100 indigenous rice genotypes, along with improved cultivars, analyzed for moisture-stress-tolerance ability, along with the checks, IR 64 Drt1 (moisture-stress tolerant) and IR 64 (moisture-stress susceptible). Among the genotypes evaluated, only 52 percent of genotypes could survive on maximum osmotic potential of (−)15 bars or 1.5 MPa on 5 DAI. The germination percent calculated for the survived rice genotypes revealed that Kuliyadichan and Rajalakshmi showed a maximum survival percentage of 88.94 and 78.89, respectively, when compared to the tolerant check, IR 64 Drt1, with a germination percent of 77.34, which is followed by the genotypes Sahbhagi Dhan (77.34%), Chandaikar (77.34%) Oheruchitteni (75.79%), Nootripathu (74.25%), Arikiraavi (71.15%), Chenkayama (71.15%), Arubathamkodai (69.61%) and Mallikar (64.96%). The results on survival percentage showed that the response of landraces significantly varied (*p* < 0.05) under a varied range of moisture stress (Figure 1a,b).

Germination rate of survived genotypes varied from 1.10 to 12.70 under moisture-stress condition (S). Among the selected moisture-stress-resilient genotypes at −1.5 MPa, Kuliyadichan (12.70) and Rajalakshmi (11.27) recorded the highest germination rate in comparison to the positive check (IR 64 Drt1) (11.05), followed by Chandaikar (11.04), Sahbhagi Dhan (11.04), Oheruchitteni (10.82), Nootripathu (10.60), Chenkayama (10.16), Arikiraavi (10.16), Arubathamkodai (9.94) and Mallikar (9.28) (Table 1).

Among the selected moisture stress-resilient genotypes, at −1.5 MPa osmotic potential (OP), Rajalakshmi and Kuliyadichan showed the maximum vigor-index values (3568.5 and 3417.5, respectively), followed by Chenkayama (2849.8), Oheruchitteni (2550.2), Chandaikar (2476.5), Sahbhagi Dhan (2162.0) and Arikiraavi (2071.5), when compared to IR 64 Drt1 (1981.2). The ten rice genotypes selected based on morphological characteristics were forwarded for further biochemical studies (Table 1).

The relative water content (RWC) of the germinating seed on 5 DAI significantly varied (*p* < 0.05) under C and S conditions. Kuliyadichan (28.05) showed the highest RWC and was on par with Rajalakshmi (27.01), Oheruchitteni (26.88), Nootripathu (25.89), Sahbhagi Dhan (25.88), Mallikar (24.99), IR 64 Drt1 (24.52) and Chenkayama (24.12). The selected rice genotypes in the investigation had a higher RWC than the susceptible check (IR 64), which had the lowest RWC value, at 10.93 (Table 1).

### 3.2. Remobilization of Seed Macromolecules in the Germinated Seeds under Moisture Stress

The starch content of the selected rice genotypes differed significantly (*p* < 0.05) under C and S conditions, influenced by α-Amylase (Table 2). The total starch content in the germinated seeds was negatively related to moisture-stress-tolerance response. Among the rice genotypes, Chandaikar (8.94 g 100 g^−1^ FW) and Kuliyadichan (10.19 g 100 g^−1^ FW) recorded the lowest starch content, compared with IR 64 Drt1 (12.89 g 100 g^−1^ FW), and are on par with each other. In the case of S, the results revealed that all of the ten selected rice genotypes recorded a lower starch content compared to the moisture-stress-susceptible check, IR 64 (34.16 g 100 g^−1^ FW) (Table 3; Figure 2a). The results showed a relatively higher remobilizing response of starch under moisture stress when compared with a controlled (C) environment.

Similarly, data on the protein content of selected rice genotypes showed significant variations (*p* < 0.05) under both the C and S environments. Remobilization of protein reserve in germinating seeds of rice genotypes under moisture stress showed an enhanced moisture-stress-tolerance capacity. Among the rice genotypes, Rajalakshmi (0.51 mg g^−1^ FW) and Nootripathu (0.51 mg g^−1^ FW) recorded the lowest mean values for protein and are close to IR 64 Drt1 (0.48 mg g^−1^ FW). The protein content in all of the genotypes decreased in stress-induced plants during seed germination, as compared to the susceptible genotype IR 64 (1006.73 mg g^−1^ FW) (Table 3 and Figure 2b).

The accumulation of total free amino acids content for selected rice genotypes varied significantly (*p* < 0.05) under C and S conditions. In stress-induced seeds, Oheruchitteni (0.472 μg g^−1^ FW) recorded the highest free amino acids content, followed by Sahbhagi Dhan (0.443 μg g^−1^ FW), Nootripathu (0.394 μg g^−1^ FW), Chenkayama (0.375 μg g^−1^ FW) and Kuliyadichan (0.370 μg g^−1^ FW), whereas the control seeds (C) showed a considerable increase in the moisture-stress-susceptible genotype, IR 64 (Figure 2c).

### 3.3. Hydrolytic Enzyme Activities under Moisture Stress

The α-amylase activity of selected rice genotypes showed a significant difference (*p* < 0.05) among C and S environments. The results implied that α-amylase activity was significantly influenced by genotypic nature (Table 2 and Table 4; Figure 3). In stress-induced treatments, Kuliyadichan (13.57 U.g^−1^) recorded the highest α-amylase activity and was on par with Chandaikar (12.95 U.g^−1^), Rajalakshmi (12.3 U.g^−1^) and IR 64 Drt1 (12.65 U.g^−1^). The moisture-stress-susceptible check (IR 64) showed the lowest α-amylase activity, at 2.13 U.g^−1^. The osmotic-stress-induced seeds of Kuliyadichan (1.19 U.g^−1^) and Rajalakshmi (1.09 U.g^−1^) recorded the highest protease activity compared with IR 64 Drt1 (1.03 U.g^−1^), followed by Mallikar (0.97 U.g^−1^), Chandaikar (0.88 U.g^−1^) and Sahbhagi Dhan (0.81 U.g^−1^). However, under the C environment, the genotypes exhibited no significant differences (Table 2 and Table 3; Figure 3). The lipase activity in Kuliyadichan (29.21 U.g^−1^) was significantly higher compared to IR 64 Drt1 (25.1 U.g^−1^) in the germinated seeds under a stressed environment. Both the treatments C and S showed significant differences (*p* < 0.05) and are highly influenced by genotypic nature. Here, also, IR 64 registered the least lipase activity under S (Table 2 and Table 3; Figure 3).

### 3.4. Endogenous Hormone Modulation in the Germinated Seeds under Moisture Stress

The variance analysis showed that moisture stress significantly influenced auxin homeostasis in the germinated seeds (*p* < 0.05) in all of the genotypes. Our experimental results exhibited a significant increase in seed endogenous IAA on 5 DAI under S when compared to the C environment. Among the rice genotypes, Kuliyadichan (58.87 µ mol g^−1^ FW) and Arubathamkodai (50.27 µ mol g^−1^ FW) recorded the highest IAA concentration compared to the moisture-stress-susceptible check, IR 64 (40.79 µ mol g^−1^ FW). More precisely, the auxin level determined in terms of IAA content in emerging rice roots showed that S pronounced a 2-fold increase of IAA in most of the genotypes (Table 4 and Figure 4).

Gibberellic acid (GA) in the seeds of all of the rice genotypes studied differed significantly (*p* < 0.05) under control and stressed environment. The mean comparison values showed that the genotype Kuliyadichan (3.18 µ mol g^−1^ FW) recorded the highest GA, followed by IR 64 Drt1 (2.350 µ mol g^−1^ FW), and this is on par with Rajalakshmi (2.28 µ mol g^−1^ FW), Sahbhagi Dhan (2.2 µ mol g^−1^ FW) and Nootripathu (2.15 µ mol g^−1^ FW). The landraces Chenkayama (0.75 µ mol g^−1^ FW) and Oheruchitteni (0.70 µ mol g^−1^ FW) recorded the lowest GA concentration, However, the GA level remarked a decreasing trend in stressed seeds in comparison with the control, irrespective of genotypes (Table 4 and Figure 4).

Data on CK content from the present study, using HPLC, revealed that moisture stress invariably reduced the CK values. The genotype Kuliyadichan proved to be superior by registering higher values (0.37 μg g^−1^), followed by Sahbhagi Dhan (0.31 μg g^−1^), under moisture stress. Out of five CKs analyzed by HPLC, the trans-zeatin production was higher when compared to others. In the present study, irrespective of the treatments hormone ABA was elevated under moisture stress (Table 4). Lower ABA production was found in IR 64 (susceptible check) under moisture stress. Kuliyadichan showed a higher ABA content, at 0.18 μg g^−1^, with a 38.4 percent increase over the respective control (C). This was followed by Mallikar, and IR 64 Drt1, with an increase of 37.5 percent over the respective irrigated plants. Rajalakshmi and IR 64DRt1 are on par with each other (Appendix A).

### 3.5. Antioxidant Enzymes Triggered under Moisture Stress

The catalase activity (CAT) in the germinating seeds of rice genotypes differed significantly (*p* < 0.05) under C and S treatments. Among the rice genotypes, the CAT activity in the genotype Rajalakshmi (2.682 mg/g) showed a higher value that is on par with Sahbhagi Dhan (2.58 U.g^−1^ FW min^−1^), as compared to the positive check, IR 64 Drt1 (2.29 U.g^−1^ FW min^−1^). Similarly, CAT activity in Kuliyadichan was on par with IR 64 Drt1, and the landrace Arubathamkodai (0.79 U.g^−1^ FW.min^−1^) showed the lowest CAT activity (Table 3 and Figure 5).

The SOD activity showed a steady decline in stress-induced seeds in a few rice genotypes (*p* < 0.05). However, Oheruchitteni (85.12 U.g^−1^ FW min^−1^), Arikiraavi (83.79 U.g^−1^ FW min^−1^) and Nootripathu (81.85 U.g^−1^ FW min^−1^) recorded the highest SOD activity. The moisture-stress-tolerant check (IR 64 Drt1) and the tolerant landrace Kuliyadichan showed a 3.07 and 3.1% reduction in SOD activity. On the contrary, Chandaikar, Nootripathu and Sahbhagi Dhan responded by showing a considerable increase in SOD activity (Table 3 and Figure 5). The results also showed a differential response pattern for SOD activity during seed germination in stress-induced seeds.

### 3.6. Remodeling Osmolytes during Germination under Moisture Stress

The accumulation of osmolytes and compatible solutes helps the plants to cope with moisture stress and maintain the osmotic turgor in cells. Accordingly, our study showed a gradual increase in proline and total soluble sugars in emerging rice seeds under moisture-deficit stress. The proline content of selected rice genotypes showed significant variation (*p* < 0.05) both under C and S conditions. Here, also, Kuliyadichan (88.21 μg g^−1^ FW) recorded the highest proline content compared with moisture-stress-tolerant check (IR 64 Drt1) (85.800 μg g^−1^ FW), followed by Rajalakshmi (81.5 μg g^−1^ FW), Sahbhagi Dhan (79.2 μg g^−1^ FW), Nootripathu (75.53 μg g^−1^ FW) and Mallikar (72.01 μg g^−1^ FW), and the landrace Chenkayama (40.2 μg g^−1^ FW) recorded the lowest proline content (Table 3; Figure 6). The total soluble sugar of selected rice genotypes also significantly varied (*p* < 0.05) among the treatments. Stress-induced seeds of Kuliyadichan (63.5 mg g^−1^ FW) recorded the highest TSS, followed by Rajalakshmi (61.00), Sahbhagi Dhan (58.60), Nootripathu (53.5 mg g^−1^ FW), Mallikar (46.2 mg g^−1^ FW), Chandaikar (42.5 mg g^−1^ FW), Arubathamkodai (35.2 mg g^−1^ FW) and Oheruchitteni (32.5 mg g^−1^ FW), compared to the susceptible genotype IR 64 (Table 3 and Figure 6).

### 3.7. Principle Component Analysis for Physiological and Metabolic Responses in Rice Genotypes during Germination Influenced by Varying Levels of Moisture Stress

The PCA was performed by using the RWC and other metabolic responses under S in 10 selected rice genotypes. The variance proportion, cumulative proportion and Eigen values are given in Appendix A). In the present study, out of eleven, three PCs exhibited Eigen values of >1.0 and 88.93% total variability among the variables studied. Among the three PCs, the first component (PC1) shared the highest total variation (66.61%), and PC2 and PC3 contributed 12.12 and 10.19% of the total variance, respectively. The Eigenvectors of the PCs for physiological and metabolic responses during seed germination under S are presented in Appendix A The coefficient values >0.3 exhibited in the overall variation were considered in order to determine the critical limit for the Eigen vectors coefficients. The results showed that TSS, GA and hydrolytic enzymes possessed the highest positive value (>0.9), followed by proline, CAT activity, RWC and IAA, in PC1. Free amino acids recorded the maximum positive value (0.91), followed by RWC (0.60), in PC2 (Appendix A).

Likewise, first the two PCs represented in the biplot showed the relation among the rice genotypes and their responses to S (Figure 7). The genotypes Nootripathu, Sahbhagi Dhan, Mallikar, Kuliyadichan and Rajalakshmi occupied the positive coordinates of the biplot. Hydrolytic enzymes, phytohormones and CAT are in the same quadrant influencing the inherent moisture-stress tolerance of these rice genotypes. Under the control condition (C), four PCs displayed >1.0 Eigen values with 88.61% total variability. Here, also, PC1 shared a high proportion of total variation, i.e., 44.6%, whereas PC2, PC3 and PC4 contributed 23.9%, 13.1% and 7.7% of the total variance, respectively. Here, in our study, we showed that lipase activity and GA had the highest positive value of 0.96, followed by proline (0.92) and protease (0.85), in PC1. However, in PC2, the highest positive value was observed in SOD, followed by free amino acids. Mallikar, Sahbhagi Dhan and Nootripathu formed a cluster in the biplot right corner, representing positive values for both PCs. In contrast to S, free amino acids, protease and lipase activity, RWC, proline and CAT activity influence the initial seed-germination process. The PCA results concluded that amylase activity, IAA and GA modulation and osmolyte accumulation coupled with antioxidant enzymes are the possible mechanisms for inherent moisture-stress tolerance and enhanced seed germination during S in rice genotypes (Figure 7; Appendix A).

## 4. Discussion

Crop improvement for moisture-stress resistance in crop plants is a challenging task, as it is a complex quantitative trait influenced by several environmental factors [44]. Landraces evolved under natural selection were the best source for improving traits controlling moisture-stress tolerance. In the present investigation, 100 rice genotypes, including 85 traditional rice landraces and 15 improved cultivars, originated from different agro-climatic zones of Tamil Nadu, India, were evaluated for their moisture-stress-tolerance potential during seed germination.

Seed germination is highly influenced by moisture stress, affecting the plant viability [45]. The germination percentage varied from 88.94 to 64.96% with the highest germination percent registered by Kuliyadichan on 5 DAI compared to moisture-stress-tolerant check (IR 64 Drt1) and susceptible check (IR 64). Only 52% of the tested rice landraces survived at the lowest osmotic potential of −1.5 MPa. In our study, the germination percentage of survived landraces marked a decreasing trend with an increase in moisture stress intensity which is in accordance with the previous studies performed with fifteen rice landraces by Gampala et al. [46]. The germination rate of the survived rice genotypes varied from 1.11 to 12.71 under induced moisture stress. Here, also, Kuliyadichan and Rajalakshmi performed far better than the moisture-stress-tolerant check (IR 64 Drt1). The results indicated that the PEG-induced moisture stress lowers the osmotic potential and impairs water availability. Seed hydration leads to the activation of key metabolic processes. Elevated moisture stress intensity reduces water uptake, thereby reducing seed germination rate and radicle development [47]. However, a few rice genotypes, namely Kuliyadichan and Rajalakshmi germinated, at (−) 1.5 MPa, suggesting that the lower osmotic potential has no impact on the seed water uptake. Previous studies reported that the landraces are in a constant state of evolution by virtue of natural and artificial selection [48]. Moreover, G × E interaction for germination rate was significant, highlighting that the landraces responded differentially to control (C) and stress (S).

The present study evidenced a significant reduction in the root and shoot length of rice genotypes during early moisture stress in comparison to control seeds (C). However, some of the genotypes exhibited higher radicle and plumule length compared to the moisture-stress-tolerant check (IR 64 Drt1). A sharp reduction in root and shoot length of the susceptible check (IR 64) under PEG-induced moisture stress is attributed to a reduction in turgor pressure, which, in turn, affects the cell elongation and expansion process [49]. The R/S ratio and vigor index are considered critical traits for identifying potential moisture-stress-tolerant genotypes [47]. The differential responses of the rice genotypes to moisture stress were related to their inherent genetic potential, and, consequently, the R/S ratio was reduced under S. Nevertheless, enhanced root growth under moisture stress is a target trait and is considered as an adaptive strategy for increased water uptake [44]. The study indicates that cultivars with a higher R/S ratio signify a good source–sink relationship and are the most preferred cultivars to screen for moisture-stress resilience [50,51]. Accordingly, Chenkayama, Oheruchitteni, Sahbhagi Dhan, Kuliyadichan, Rajalakshmi, Chandaikar and Nootripathu showed a significant R/S ratio compared to IR 64 (Drt1), and this warranted a better source–sink relationship. Furthermore, the RWC of rice genotypes declined under S compared with C, wherein few landraces registered a higher RWC in comparison with IR 64 Drt1. The findings suggest a wide spectrum of variation among the rice genotypes for their sensitivity to moisture stress. The traditional landraces may have improved mechanisms for cellular osmotic adjustments to preserve membrane damage and sustain turgidity in stressed environments [52]. Hence, the selected rice genotypes can maintain water in the cell in comparison to their counterpart, IR 64, which is moisture-stress sensitive.

Starch degradation under stress is a common plant response and contributes to sugar accumulation under stress conditions [53]. Sugars produced by starch remobilization act as signaling cues, and they crosstalk with the ABA-dependent signaling pathway to activate downstream components involving stress response cascade [54]. Gonzalez-Cruz and Pastenes, [55] found that a moisture-stress-resistant variety of broad bean (*Phaseolus vulgaris*) degraded more starch than a moisture-stress-sensitive variety. Accordingly, our results showed that hydrolysis of starch had a positive relation with moisture-stress tolerance in rice genotypes (Kuliyadichan, Chandaikar, Mallikar, Nootripathu, Rajalakshmi, Sahbhagi Dhan, Arubathamkodai, Arikiraavi, Chenkayama and Oheruchitteni) compared to IR 64 Drt1. The remobilization of starch is slow in the susceptible check (IR 64), with the maximum starch content under S, whereas it is the opposite under a non-stressed environment. Solutes released during storage proteins catalysis contribute to seed germination and initiate radicle protrusion [56]. Our study concluded that the genotypes Kuliyadichan, Chandaikar, Mallikar, Nootripathu, Rajalakshmi, Sahbhagi Dhan and Arubathamkodai showed a significant reduction in protein content (26.55 to 13.66%) compared to IR 64 (susceptible check). Compatible solutes, such as amino acids, serve a function in cells to lower or balance the osmotic potential of intracellular and extracellular ions in response to osmotic stresses [57]. In our study, Kuliyadichan, Chandaikar, Mallikar, Nootripathu, Rajalakshmi, Sahbhagi Dhan, Arubathamkodai, Arikiraavi, Chenkayama and Oheruchitteni showed increased amino acid accumulation (13 to 116% increase over control) under moisture-stress conditions. Similar results of increased total free amino acids accumulation in seven fab varieties under water stress were recorded [58]. The dynamic changes in the interconversion of starch and protein within seed source and sink promote better abiotic-stress tolerance [19].

Seed imbibition activates hydrolytic enzymes during germination and mobilizes the seed macromolecules in the endosperm to simpler molecules. Hydrolytic enzymes are associated with water activity. Stress causes a significant reduction in water activity and thereby exerts a negative effect in activating hydrolases [59]. Likewise, the present study showed a significant reduction in amylase activity under S when compared to C in rice genotypes on 5 DAI. However, the genotypes Kuliyadichan, Nootripathu, Chandaikar, Sahbhagi Dhan, Rajalakshmi and Arubathamkodai performed on par with IR 64 Drt1, suggesting their genetic architect to withstand less water activity. The amylase activity in the cotyledons increased and reached maximum on fifth day of germination and depends on steeping factors, while the soluble sugars increased and starch decreased [60]. An impaired amylase biosynthetic mechanism caused by S prevents the germination process, as in the case of the sensitive genotype IR 64. On the contrary, studies in *Agropyronde sertorum* seeds revealed that GAs alleviate the S effects caused by α-amylase synthesis inhibition [61]. Proteases are synthesized (as inactive or active precursors) and activated downstream by other processing enzymes or autocatalytically [62]. Cysteine proteases are the predominant protease group among rice and other cereals [14]. The results of the present study witnessed a reduction in protease activity under S and significant G × T interaction noticed. The landraces Kuliyadichan and Rajalakshmi recorded maximum proteolytic activity and confirm efficient protein mobilization, which supports further protein biosynthesis in embryo and endosperm. Similarly, lipase activity under S showed significant G × T interaction, highlighting that the rice genotypes responded differentially to the environments. Maximum lipase activity was observed under control where the water activity is high. The triglycerides stored in oleosomes are hydrolyzed by lipases to energy, which provides carbon backbones for embryonic growth. Moisture stress impedes gluconeogenesis and sugar synthesis, thereby affecting carbon transport in germinating seeds. This, in turn, inhibits seed germination. Our study identified a few rice genotypes with significant lipase activity, such as Kuliyadichan, Nootripathu, Chandaikar, Sahbhagi Dhan and Rajalakshmi, in contrast with IR 64 Drt1.

Modifications in the synthesis, transport and signaling of auxin profoundly affect moisture-stress resistance in rice. An increase in auxin level due to the overexpression of auxin efflux carrier gene *OsGH3.2* [63] resulted in improved moisture-stress tolerance through the regulation of root architecture, ABA-responsive genes expression, ROS metabolism and metabolic homeostasis [64]; under moisture-stress condition, the auxin response was significantly modulated. In the present study, the genotypes Kuliyadichan, Chandaikar, Mallikar, Nootripathu, Rajalakshmi, Sahbhagi Dhan, Arubathamkodai, Arikiraavi, Chenkayama and Oheruchitteni showed increased IAA under moisture stress compared to that of the control condition (C). Auxins—more specifically, IAA—regulate the seed-germination process in a crosstalk with gibberellic acid (GA), abscisic acid (ABA) and ethylene (ET) signaling pathways [65,66,67]. Furthermore, there exists an antagonistic relationship between ABA and GA. Seed imbibition activates GA biosynthesis, which, in turn, elicits downstream endosperm hydrolyzing enzymes, thereby alleviating the inhibitory effects of ABA on embryo development. However, under S, high ABA and low GA levels inhibit water uptake, which, in turn, prevents cell-wall loosening and reduces embryo growth [68]. Our study revealed that, during S, the rice genotypes showed a remarkable increase in endogenous GA level over the tolerant check, IR 64 Drt1; thus, it might be their inherent genetic makeup to adapt to an unfavorable environment. The reason might be due to the repressive effect of GA on ABA during early phases of germination through eliciting the expression of genes encoding cell-wall-loosening enzymes, such as α-expansins [69]. Thus, GA exhibits interplay between hydrolytic enzymes and cysteine proteases in plant-hormone-mediated stress responses that is pronounced in the traditional rice landraces.

Seed germination directly depends on ROS homeostasis that triggers associated cellular events. However, under a stressed environment, ROS accumulation inhibits seed germination and aligns with the oxidative window for seed germination [70]. ROS moieties generated during seed germination are superoxide, hydrogen peroxide and free hydroxyl radicals and are regulated by enzymatic and non-enzymatic ROS scavenging mechanisms [71]. Significant changes of enzymatic antioxidants, such as superoxide dismutase, catalase and ascorbate-glutathione, during seed germination were previously reported [72,73]. Accordingly, the present study warrants activation of enzymatic antioxidants, CAT and SOD in response to S. The rice genotypes exhibited increased catalase activity (2.4 to 54% increase over control), whereas SOD varied with genotypes. Interestingly, genotypes with higher SOD activities suggest more oxidative stress and O_2_^−^ generation in response to S. In order to maintain the ROS homeostasis, SOD would be activated in few rice genotypes under study [74]. Thus, the genotypes possess an inherent potential for early activation of antioxidant enzymes to prevent excess oxidative damage and ROS accumulation. The O_2_^−^ generation in the rest of the landraces might be below the critical limit and is more relevant in hormonal signaling [75]. Further, the correlation between ROS and phytohormones, such as ABA and GA, were well established by earlier studies. ROS accumulation remodels ABA synthesis, which is a germination repressor and an antagonist to GA, as discussed in the previous section. Increased CAT activity in the rice landraces indicates that H_2_O_2_ accumulation in germinating seeds degrades ABA via activation of ABA-8-hydroxylase and regulates its interaction with GA [72,76]. H_2_O_2_ stimulates GA/ABA balance in favor of GA, which, in turn, triggers seed germination during stress [77,78].

The genotypes showed significant variation (*p* < 0.01) with respect to osmolyte accumulation (proline and sugar). The tolerant check (IR 64 Drt1), along with Kuliyadichan, Sahbhagi Dhan and Rajalakshmi, showed higher proline and sugar accumulation under S compared to the susceptible check (IR 64). In general, moisture-stress tolerance mainly depends on the osmotic adjustment and maintenance of seed turgor [79]. The result of the present study indicates the prospects of rice genotypes for the regulation of redox potential and osmoprotection under S [80]. Hence, the study authenticated that a higher accumulation of osmolytes augments better protection from oxidative stress during seed germination, which is a feature of traditional rice landraces.

The correlation coefficients between morpho-physiological, biochemical traits, metabolic responses and germination rate during seed germination in different rice genotypes under S (−1.5 Mpa) (Figure 8a; Appendix A) revealed that the germination rate (GR) exhibited significant negative correlation with starch and SOD. However, positive correlations between GR and α-amylase, protease, lipase, catalase, TSS and RWC were observed. More precisely, GA showed positive correlation with hydrolytic enzymes and a significant negative correlation with starch and protein content under S. Likewise, IAA was strongly correlated with α-amylase and protease and indirectly proportionate to starch content. The results put forth that, under S conditions, the remobilization of seed reserves through hydrolytic enzymes is an intriguing factor in regulating phytohormone synthesis and ROS scavenging enzymes. However, under C, GR was significantly correlated with hydrolytic enzymes, GA and osmolytes (sugar and proline) (Figure 8b; Appendix A). Previous studies suggest that early seedling vigor under moisture stress was positively correlated with RWC and accumulation of osmolytes [52,81]. Our findings suggested that the germination rate and seedling survival under S were attributed to phytohormone modulation (Appendix A), enzymatic antioxidants, triggered hydrolytic enzyme activities and osmoprotection.

## 5. Conclusions

The present investigation aimed at identifying potential local donor candidates toward genetic improvement of moisture-stress tolerance in rice. The study concluded six rice genotypes, namely Kuliyadichan, Rajalakshmi, Sahbhagi Dhan, Nootripathu, Chandaikar and Mallikar, selected among 100 rice genotypes representing different agro-climatic zones of Tamil Nadu. Deciphering the possible mechanisms for moisture-stress tolerance revealed that the germination rate and vigor index were governed by a synergistic modulation of phytohormones, antioxidants, hydrolytic enzymes and osmolyte accumulation. Further studies using omic approaches will give more understanding of the metabolic pathways and genes activated in these rice genotypes in contrast with moisture-stress-tolerant (IR 64 Drt1).

## Figures and Tables

**Figure 1 plants-11-00775-f001:**
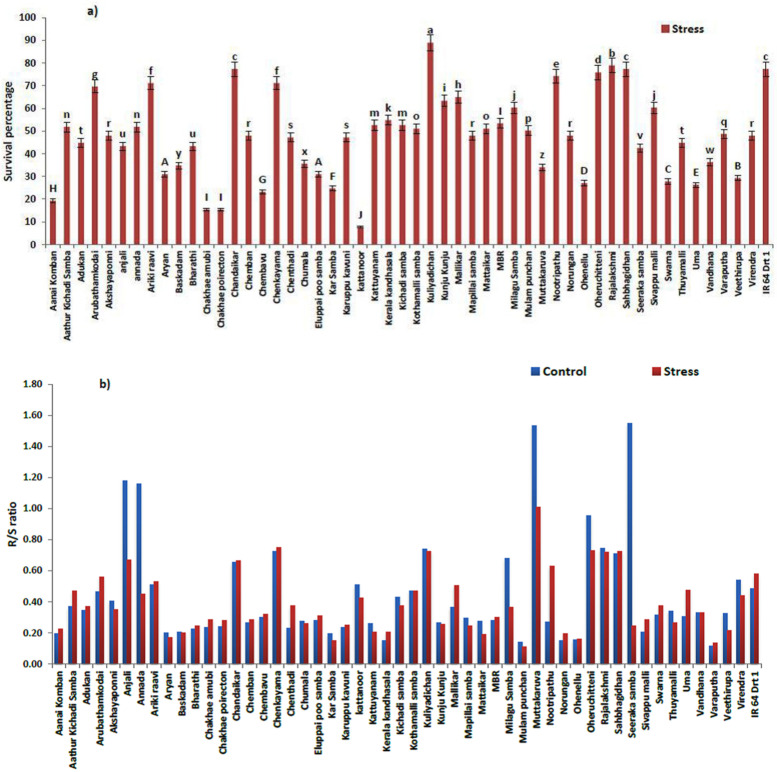
(**a**) Survival percent. (**b**) R/S ratio of rice genotypes under moisture stress (−1.5 MPa). Data represented are mean (±standard error) (*n* = 3), and values followed by the same lowercase letter are not significantly different from each other, as determined by DMRT (*p* ≤ 0.05). IR 64 and IR 64 Drt1 used as negative and positive checks, respectively.

**Figure 2 plants-11-00775-f002:**
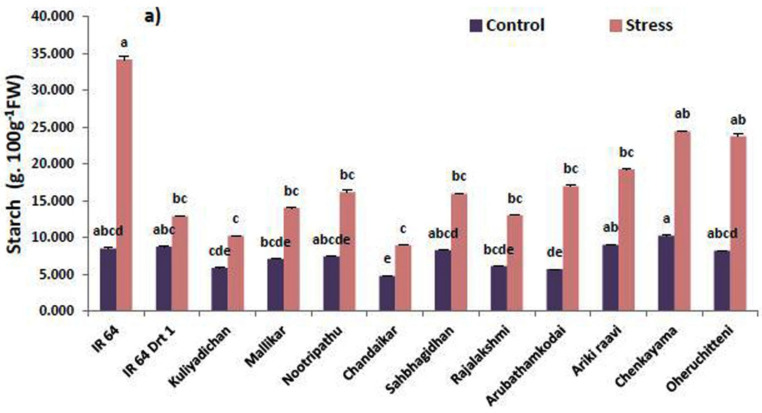
Influence of moisture stress on (**a**) starch, (**b**) protein and (**c**) amino acid content in the germinated seeds on 5 DAI in selected rice genotypes. Data represented are mean (±standard error) (*n* = 3), and values followed by the same lowercase letter are not significantly different from each other, as determined by DMRT (*p* ≤ 0.05). IR 64 and IR 64 (Drt1) used as negative and positive check, respectively, and stress imposed at −1.5 MPa.

**Figure 3 plants-11-00775-f003:**
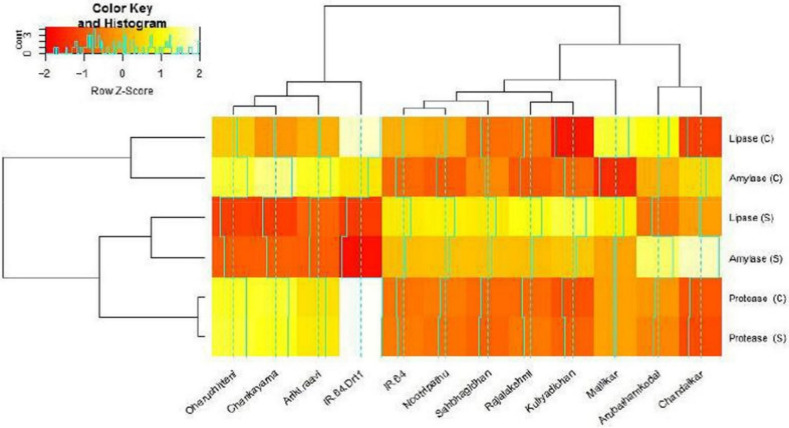
Heatmap and clustering of selected rice genotypes based on hydrolytic enzyme activities, such as α-amylases, proteases and lipases. NS—non-stressed environment; C—controlled environment; S—induced moisture stress of −1.5 MPa in comparison to IR 64 (Drt1).

**Figure 4 plants-11-00775-f004:**
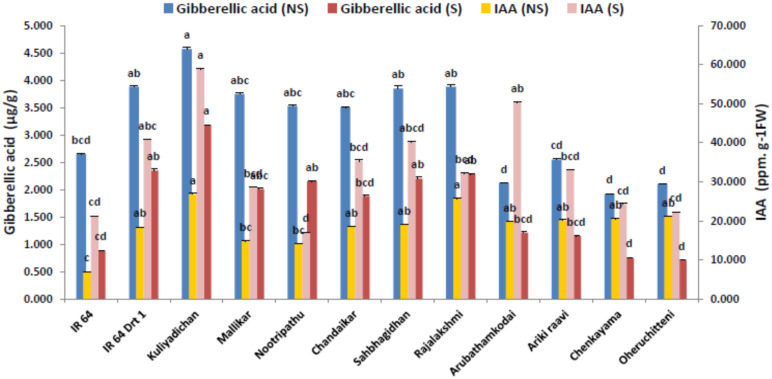
Influence of moisture stress on phytohormones, GA and IAA in the germinated seeds on 5 DAI in selected rice genotypes. Data represented are mean (±standard error) (*n* = 3), and values followed by the same lowercase letter are not significantly different from each other, as determined by DMRT (*p* ≤ 0.05). IR 64 and IR 64 (Drt1) used as negative and positive check, respectively, and stress imposed at −1.5 MPa.

**Figure 5 plants-11-00775-f005:**
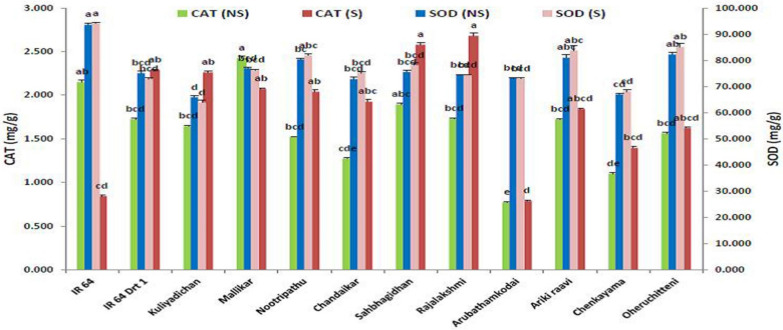
Influence of moisture stress on antioxidants, CAT and SOD in the germinated seed on 5 DAI in selected rice genotypes. Data represented are mean (±standard error) (*n* = 3), and values followed by the same lowercase letter are not significantly different from each other, as determined by DMRT (*p* ≤ 0.05). IR 64 and IR 64 Drt1 used as negative and positive check, respectively, and stress imposed at −1.5 MPa.

**Figure 6 plants-11-00775-f006:**
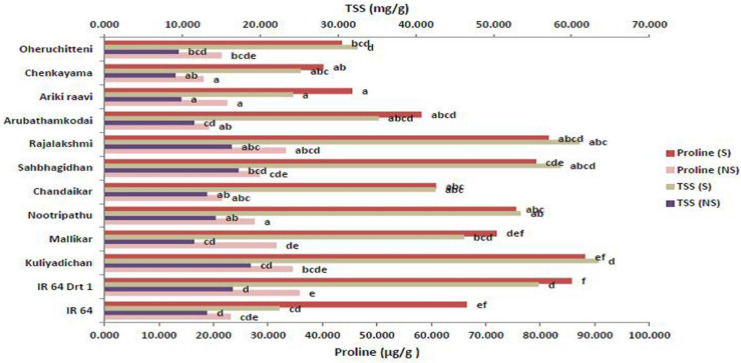
Influence of moisture stress on osmolytes, proline and sugars during seed germination on 5 DAI in selected rice genotypes. Data represented are mean (±standard error) (*n* = 3), and values followed by the same lowercase letter are not significantly different from each other, as determined by DMRT (*p* ≤ 0.05). IR 64 and IR 64 Drt1 used as negative and positive check, respectively, and stress imposed at −1.5 MPa.

**Figure 7 plants-11-00775-f007:**
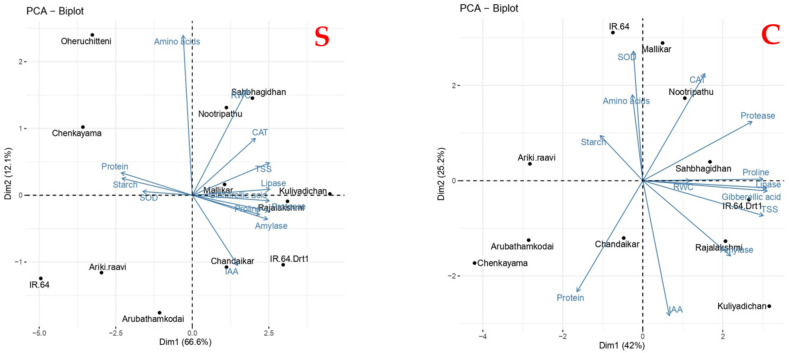
Biplots of principal component analysis under osmotic stress (−1.5 Mpa) for morpho-physiological, biochemical and metabolic responses in rice genotypes: S—induced moisture stress of −1.5 MPa in comparison to IR 64 Drt1; C—controlled environment.

**Figure 8 plants-11-00775-f008:**
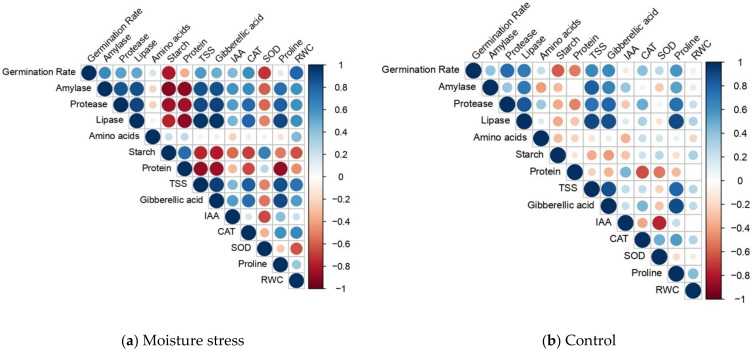
Correlation between morpho-physiological characteristics, phytohormone modulation and antioxidant enzymes in response to induced oxidative stress in selected rice genotypes: (**a**) S—induced moisture stress of −1.5 MPa; (**b**) C—controlled environment).

**Table 1 plants-11-00775-t001:** Comparison of mean values between osmotic stress (−1.5 MPa) and non-stressed (C) traditional rice genotypes.

Genotypes	Stress Intensity	Germination Rate	Root Length	Shoot Length	RS	Vigor Index	RWC
Aanai Komban	C	3.08 ± 0.03	4.66 ± 0.08	24.02 ± 0.44	0.194 ± 0.001	2334.44 ± 40.00	38.13 ± 0.08
	−1.5 MPa	2.76 ± 0.11	4.00 ± 0.07	17.66 ± 0.32	0.227 ± 0.001	423.56 ± 18.59	23.15 ± 0.10
Aathur Kichadi Samba	C	3.67 ± 0.03	12.67 ± 0.21	34.39 ± 0.63	0.368 ± 0.002	4554.98 ± 78.05	33.54 ± 0.07
	−1.5 MPa	7.40 ± 0.30	10.64 ± 0.18	22.68 ± 0.41	0.469 ± 0.002	1743.52 ± 76.52	22.64 ± 0.11
Adukan	C	3.62 ± 0.03	8.31 ± 0.14	24.22 ± 0.45	0.343 ± 0.002	3106.13 ± 53.22	38.13 ± 0.08
	−1.5 MPa	6.41 ± 0.26	5.47 ± 0.09	14.85 ± 0.27	0.369 ± 0.001	921.09 ± 40.42	23.15 ± 0.10
Arubathamkodai	C	3.63 ± 0.03	7.60 ± 0.13	16.25 ± 0.30	0.468 ± 0.002	2282.62 ± 39.11	34.50 ± 0.08
	−1.5 MPa	9.94 ± 0.41	6.18 ± 0.10	11.04 ± 0.20	0.560 ± 0.002	1209.94 ± 53.10	20.50 ± 0.11
Akshayaponni	C	3.67 ± 0.03	10.13 ± 0.17	25.02 ± 0.46	0.405 ± 0.002	3401.70 ± 58.29	44.15 ± 0.06
	−1.5 MPa	6.85 ± 0.28	7.30 ± 0.12	20.87 ± 0.38	0.350 ± 0.001	1364.81 ± 59.90	22.13 ± 0.13
Anjali	C	3.63 ± 0.03	9.69 ± 0.16	8.22 ± 0.15	1.179 ± 0.006	1707.52 ± 29.26	34.35 ± 0.10
	−1.5 MPa	6.19 ± 0.25	5.98 ± 0.10	8.93 ± 0.16	0.670 ± 0.002	651.59 ± 28.60	12.69 ± 0.19
Annada	C	3.67 ± 0.03	14.06 ± 0.24	12.14 ± 0.22	1.159 ± 0.006	2525.09 ± 43.27	38.13 ± 0.08
	−1.5 MPa	7.40 ± 0.30	5.67 ± 0.10	12.64 ± 0.23	0.449 ± 0.002	958.67 ± 42.07	23.15 ± 0.10
Arikiraavi	C	3.08 ± 0.03	11.35 ± 0.19	22.23 ± 0.41	0.511 ± 0.003	2724.87 ± 46.69	34.14 ± 0.09
	−1.5 MPa	10.17 ± 0.41	9.97 ± 0.17	18.86 ± 0.35	0.529 ± 0.002	2071.50 ± 90.91	14.59 ± 0.12
Aryan	C	3.08 ± 0.03	8.71 ± 0.15	43.06 ± 0.79	0.202 ± 0.001	4213.39 ± 72.20	32.25 ± 0.10
	−1.5 MPa	4.42 ± 0.18	4.76 ± 0.08	27.99 ± 0.51	0.170 ± 0.001	1025.18 ± 44.99	20.24 ± 0.14
Baskadam	C	3.67 ± 0.03	5.07 ± 0.09	24.92 ± 0.46	0.203 ± 0.001	2907.44 ± 49.82	25.31 ± 0.09
	−1.5 MPa	4.97 ± 0.20	3.55 ± 0.06	17.76 ± 0.32	0.200 ± 0.001	750.02 ± 32.92	10.22 ± 0.09
Bharathi	C	3.08 ± 0.03	5.98 ± 0.10	26.41 ± 0.49	0.226 ± 0.001	2635.40 ± 45.16	38.13 ± 0.08
	−1.5 MPa	6.19 ± 0.25	3.95 ± 0.07	16.15 ± 0.30	0.245 ± 0.001	880.53 ± 38.64	23.15 ± 0.10
Chakhaeamubi	C	3.08 ± 0.03	8.61 ± 0.15	36.18 ± 0.67	0.238 ± 0.001	3644.01 ± 62.44	33.78 ± 0.10
	−1.5 MPa	2.21 ± 0.09	6.49 ± 0.11	22.88 ± 0.42	0.284 ± 0.001	459.13 ± 20.15	23.16 ± 0.11
Chakhaepoirecton	C	3.08 ± 0.03	8.31 ± 0.14	34.78 ± 0.64	0.239 ± 0.001	3505.73 ± 60.07	31.49 ± 0.12
	−1.5 MPa	2.21 ± 0.09	6.69 ± 0.11	23.98 ± 0.44	0.279 ± 0.001	479.57 ± 21.05	21.86 ± 0.12
Chandaikar	C	3.68 ± 0.03	16.52 ± 0.28	25.22 ± 0.46	0.655 ± 0.003	4045.60 ± 69.32	38.13 ± 0.08
	−1.5 MPa	11.05 ± 0.45	12.67 ± 0.21	19.06 ± 0.35	0.665 ± 0.002	2476.48 ± 108.69	23.15 ± 0.10
Chemban	C	3.08 ± 0.03	10.34 ± 0.17	38.67 ± 0.71	0.267 ± 0.001	3985.64 ± 68.29	34.50 ± 0.08
	−1.5 MPa	6.85 ± 0.28	6.59 ± 0.11	22.88 ± 0.42	0.288 ± 0.001	1428.18 ± 62.68	20.50 ± 0.11
Chembavu	C	3.08 ± 0.03	9.93 ± 0.17	33.29 ± 0.61	0.298 ± 0.002	3513.87 ± 60.21	33.54 ± 0.07
	−1.5 MPa	3.31 ± 0.14	6.49 ± 0.11	20.17 ± 0.37	0.322 ± 0.001	625.02 ± 27.43	22.64 ± 0.11
Chenkayama	C	2.31 ± 0.02	18.54 ± 0.31	25.61 ± 0.47	0.724 ± 0.004	2684.20 ± 45.99	54.55 ± 0.67
	−1.5 MPa	10.17 ± 0.41	17.02 ± 0.29	22.68 ± 0.41	0.751 ± 0.003	2849.76 ± 125.07	24.13 ± 0.66
Chenthadi	C	3.08 ± 0.03	8.51 ± 0.14	37.08 ± 0.68	0.230 ± 0.001	3709.08 ± 63.55	38.13 ± 0.08
	−1.5 MPa	6.74 ± 0.27	7.50 ± 0.13	19.97 ± 0.37	0.376 ± 0.001	1309.23 ± 57.46	23.15 ± 0.10
Chumala	C	2.31 ± 0.02	7.50 ± 0.13	27.11 ± 0.50	0.277 ± 0.001	2110.76 ± 36.17	30.18 ± 0.07
	−1.5 MPa	5.08 ± 0.21	5.27 ± 0.09	20.37 ± 0.37	0.259 ± 0.001	922.19 ± 40.47	17.35 ± 0.12
Eluppai Poo Samba	C	3.08 ± 0.03	10.64 ± 0.18	37.87 ± 0.70	0.281 ± 0.001	3944.97 ± 67.60	28.69 ± 0.07
	−1.5 MPa	4.42 ± 0.18	7.50 ± 0.13	24.28 ± 0.44	0.309 ± 0.001	993.74 ± 43.61	8.44 ± 0.11
Kar Samba	C	2.31 ± 0.02	6.59 ± 0.11	33.29 ± 0.61	0.198 ± 0.001	2434.08 ± 41.71	46.99 ± 0.09
	−1.5 MPa	3.54 ± 0.14	3.14 ± 0.05	20.77 ± 0.38	0.151 ± 0.001	598.76 ± 26.28	13.97 ± 0.15
Karuppukavuni	C	3.77 ± 0.03	9.83 ± 0.17	41.86 ± 0.77	0.235 ± 0.001	5152.54 ± 88.29	47.20 ± 0.07
	−1.5 MPa	6.74 ± 0.27	7.19 ± 0.12	28.90 ± 0.53	0.249 ± 0.001	1721.66 ± 75.56	10.93 ± 0.11
Kattanoor	C	3.77 ± 0.03	6.08 ± 0.10	11.94 ± 0.22	0.509 ± 0.003	1788.61 ± 30.65	47.20 ± 0.07
	−1.5 MPa	1.10 ± 0.05	4.35 ± 0.07	10.23 ± 0.19	0.425 ± 0.001	113.89 ± 5.00	10.93 ± 0.11
Kattuyanam	C	3.77 ± 0.03	5.17 ± 0.09	19.83 ± 0.37	0.261 ± 0.001	2487.18 ± 42.62	38.13 ± 0.08
	−1.5 MPa	7.51 ± 0.31	3.45 ± 0.06	16.86 ± 0.31	0.204 ± 0.001	1079.90 ± 47.39	23.15 ± 0.10
Kerala Kandhasala	C	3.77 ± 0.03	5.27 ± 0.09	34.39 ± 0.63	0.153 ± 0.001	3956.59 ± 67.80	24.84 ± 0.09
	−1.5 MPa	7.84 ± 0.32	4.36 ± 0.07	21.27 ± 0.39	0.205 ± 0.001	1423.38 ± 62.47	12.28 ± 0.08
Kichadi Samba	C	3.08 ± 0.03	10.84 ± 0.18	25.22 ± 0.46	0.430 ± 0.002	2928.22 ± 50.17	38.13 ± 0.08
	−1.5 MPa	7.51 ± 0.31	6.79 ± 0.11	18.06 ± 0.33	0.376 ± 0.001	1320.47 ± 57.95	23.15 ± 0.10
Kothamalli Samba	C	3.08 ± 0.03	10.54 ± 0.18	22.43 ± 0.41	0.470 ± 0.002	2676.07 ± 45.85	24.12 ± 0.10
	−1.5 MPa	7.29 ± 0.30	7.19 ± 0.12	15.25 ± 0.28	0.472 ± 0.002	1157.10 ± 50.78	10.96 ± 0.14
Kuliyadichan	C	3.82 ± 0.01	18.75 ± 0.32	25.42 ± 0.47	0.738 ± 0.004	4440.41 ± 54.40	51.70 ± 0.38
	−1.5 MPa	12.71 ± 0.52	16.01 ± 0.27	22.07 ± 0.40	0.725 ± 0.003	3417.54 ± 149.99	28.05 ± 0.49
Kunju Kunju	C	3.78 ± 0.03	10.74 ± 0.18	40.46 ± 0.75	0.266 ± 0.001	5116.55 ± 87.67	28.69 ± 0.07
	−1.5 MPa	9.06 ± 0.37	7.30 ± 0.12	28.49 ± 0.52	0.256 ± 0.001	2295.03 ± 100.72	8.44 ± 0.11
Mallikar	C	3.78 ± 0.03	7.50 ± 0.13	20.43 ± 0.38	0.367 ± 0.002	2744.17 ± 26.88	47.20 ± 0.07
	−1.5 MPa	9.28 ± 0.38	6.59 ± 0.11	13.04 ± 0.24	0.505 ± 0.002	1277.11 ± 63.99	25.00 ± 0.73
Mapillai Samba	C	3.58 ± 0.03	10.44 ± 0.18	35.38 ± 0.65	0.295 ± 0.002	4331.17 ± 74.21	24.84 ± 0.09
	−1.5 MPa	6.85 ± 0.28	6.49 ± 0.11	26.49 ± 0.48	0.245 ± 0.001	1598.78 ± 70.17	12.28 ± 0.08
Mattaikar	C	3.74 ± 0.03	7.50 ± 0.13	27.01 ± 0.50	0.278 ± 0.001	3412.97 ± 58.48	42.67 ± 0.07
	−1.5 MPa	7.29 ± 0.30	3.85 ± 0.07	20.17 ± 0.37	0.191 ± 0.001	1240.13 ± 54.43	10.63 ± 0.17
MBR	C	3.58 ± 0.03	9.22 ± 0.16	32.69 ± 0.60	0.282 ± 0.001	3962.36 ± 67.89	24.84 ± 0.09
	−1.5 MPa	7.62 ± 0.31	6.59 ± 0.11	21.97 ± 0.40	0.300 ± 0.001	1540.60 ± 67.61	12.28 ± 0.08
Milagu Samba	C	3.78 ± 0.03	15.71 ± 0.27	23.12 ± 0.43	0.679 ± 0.004	3867.39 ± 66.27	25.98 ± 0.07
	−1.5 MPa	8.62 ± 0.35	8.01 ± 0.14	21.97 ± 0.40	0.364 ± 0.001	1827.40 ± 80.20	8.06 ± 0.08
Mulampunchan	C	3.58 ± 0.03	3.37 ± 0.06	23.71 ± 0.44	0.142 ± 0.001	578.26 ± 9.91	28.46 ± 0.06
	−1.5 MPa	7.18 ± 0.29	2.43 ± 0.04	21.87 ± 0.40	0.111 ± 0.000	1236.67 ± 54.27	10.57 ± 0.09
Muttakaruva	C	3.77 ± 0.03	12.95 ± 0.22	8.44 ± 0.16	1.535 ± 0.008	2113.39 ± 36.21	24.84 ± 0.09
	−1.5 MPa	4.86 ± 0.20	6.89 ± 0.12	6.82 ± 0.12	1.011 ± 0.004	470.23 ± 20.64	12.28 ± 0.08
Nootripathu	C	3.67 ± 0.03	5.88 ± 0.10	21.63 ± 0.40	0.272 ± 0.001	2620.77 ± 26.16	47.20 ± 0.07
	−1.5 MPa	10.61 ± 0.43	5.88 ± 0.10	9.33 ± 0.17	0.630 ± 0.002	1130.82 ± 56.72	25.90 ± 0.21
Norungan	C	3.77 ± 0.03	4.76 ± 0.08	31.40 ± 0.58	0.152 ± 0.001	3601.44 ± 61.71	24.84 ± 0.09
	−1.5 MPa	6.85 ± 0.28	3.45 ± 0.06	17.66 ± 0.32	0.195 ± 0.001	1023.61 ± 44.92	12.28 ± 0.08
Ohenellu	C	3.77 ± 0.03	4.97 ± 0.08	31.69 ± 0.58	0.157 ± 0.001	3651.18 ± 62.56	38.13 ± 0.08
	−1.5 MPa	3.87 ± 0.16	3.95 ± 0.07	24.28 ± 0.44	0.163 ± 0.001	773.21 ± 33.93	23.15 ± 0.10
Oheruchitteni	C	3.58 ± 0.03	20.17 ± 0.34	21.13 ± 0.39	0.955 ± 0.005	3886.71 ± 66.60	45.44 ± 0.62
	−1.5 MPa	10.83 ± 0.44	14.09 ± 0.24	19.26 ± 0.35	0.731 ± 0.003	2550.22 ± 111.92	26.89 ± 0.28
Rajalakshmi	C	3.78 ± 0.03	20.98 ± 0.35	28.21 ± 0.52	0.744 ± 0.004	4896.70 ± 83.90	49.67 ± 2.04
	−1.5 MPa	11.27 ± 0.46	18.75 ± 0.32	26.09 ± 0.48	0.719 ± 0.003	3568.48 ± 156.61	27.01 ± 0.20
Sahbhagi Dhan	C	3.77 ± 0.03	13.68 ± 0.23	19.34 ± 0.36	0.708 ± 0.004	3273.13 ± 56.08	41.61 ± 1.33
	−1.5 MPa	11.05 ± 0.45	11.65 ± 0.20	16.05 ± 0.29	0.726 ± 0.003	2162.00 ± 94.88	25.89 ± 0.33
Seeraka Samba	C	3.58 ± 0.03	3.55 ± 0.06	2.29 ± 0.04	1.548 ± 0.008	548.49 ± 9.40	35.73 ± 0.07
	−1.5 MPa	6.08 ± 0.25	2.63 ± 0.04	10.64 ± 0.19	0.248 ± 0.001	570.77 ± 25.05	19.95 ± 0.12
Sivappumalli	C	3.80 ± 0.03	5.57 ± 0.09	27.41 ± 0.50	0.203 ± 0.001	3314.27 ± 56.79	39.07 ± 0.06
	−1.5 MPa	8.62 ± 0.35	4.36 ± 0.07	15.25 ± 0.28	0.286 ± 0.001	1195.78 ± 52.48	12.94 ± 0.09
Swarna	C	3.77 ± 0.03	9.22 ± 0.16	29.20 ± 0.54	0.316 ± 0.002	3758.32 ± 37.18	47.20 ± 0.07
	−1.5 MPa	3.98 ± 0.16	8.31 ± 0.14	22.07 ± 0.40	0.377 ± 0.001	847.11 ± 42.38	10.93 ± 0.11
Thuyamalli	C	3.77 ± 0.03	6.99 ± 0.12	20.43 ± 0.38	0.342 ± 0.002	2725.95 ± 46.71	39.07 ± 0.06
	−1.5 MPa	6.41 ± 0.26	4.36 ± 0.07	16.25 ± 0.30	0.268 ± 0.001	934.77 ± 41.02	12.94 ± 0.09
Uma	C	3.58 ± 0.03	12.57 ± 0.21	41.06 ± 0.76	0.306 ± 0.002	4986.09 ± 49.42	41.26 ± 0.09
	−1.5 MPa	3.76 ± 0.15	11.96 ± 0.20	25.08 ± 0.46	0.477 ± 0.002	975.40 ± 48.86	7.00 ± 0.16
Vandhana	C	3.77 ± 0.03	6.18 ± 0.10	18.84 ± 0.35	0.328 ± 0.002	2487.18 ± 42.62	35.77 ± 0.09
	−1.5 MPa	5.19 ± 0.21	4.46 ± 0.08	13.55 ± 0.25	0.329 ± 0.001	661.42 ± 29.03	14.91 ± 0.13
Varaputha	C	2.31 ± 0.02	4.66 ± 0.08	39.37 ± 0.73	0.118 ± 0.001	2690.30 ± 46.10	28.95 ± 0.09
	−1.5 MPa	6.96 ± 0.28	3.65 ± 0.06	26.49 ± 0.48	0.138 ± 0.000	1485.89 ± 65.21	9.91 ± 0.14
Veethirupa	C	3.77 ± 0.03	7.60 ± 0.13	23.42 ± 0.43	0.325 ± 0.002	3084.10 ± 52.85	28.69 ± 0.07
	−1.5 MPa	4.20 ± 0.17	4.15 ± 0.07	19.46 ± 0.36	0.213 ± 0.001	702.06 ± 30.81	8.44 ± 0.11
Virendra	C	3.55 ± 0.03	15.61 ± 0.26	29.00 ± 0.53	0.538 ± 0.003	4178.39 ± 71.60	36.27 ± 0.08
	−1.5 MPa	6.85 ± 0.28	8.51 ± 0.14	19.26 ± 0.35	0.442 ± 0.002	1345.32 ± 59.04	19.10 ± 0.12
IR 64 Drt1	C	3.77 ± 0.03	10.94 ± 0.18	22.62 ± 0.42	0.484 ± 0.003	3332.80 ± 57.11	50.68 ± 2.35
	−1.5 MPa	11.05 ± 0.45	9.32 ± 0.16	16.05 ± 0.29	0.581 ± 0.002	1981.18 ± 86.95	24.52 ± 0.53

Values represented are mean of three replications (±standard error).

**Table 2 plants-11-00775-t002:** Influence of moisture stress on hydrolytic enzyme activities in the germinated seeds in selected rice genotypes.

Germplasm	α-Amylase (U.g^−1^ FW)		Protease (U.g^−1^ FW)		Lipase (U.g^−1^ FW)	
	C	S	C	S	C	S
IR 64	19.55 ^bcde^	2.13 ^d^	1.72 ^ab^	0.33 ^f^	29.73 ^cde^	5.51 ^d^
IR 64 Drt1	23.72 ^ab^	12.65 ^a^	1.87 ^a^	1.03 ^abc^	40.07 ^a^	25.10 ^ab^
Kuliyadichan	24.51 ^a^	13.57 ^a^	1.71 ^ab^	1.19 ^a^	38.51 ^ab^	29.21 ^a^
Mallikar	15.33 ^e^	8.70 ^abcd^	1.88 ^a^	0.97 ^abcd^	31.30 ^bcde^	18.88 ^abcd^
Nootripathu	20.97 ^abcd^	11.50 ^ab^	1.83 ^a^	0.66 ^bcdef^	36.33 ^abc^	23.42 ^ab^
Chandaikar	22.34 ^abc^	12.95 ^a^	1.67 ^ab^	0.88 ^abcde^	29.20 ^cdef^	15.20 ^abcd^
Sahbhagi Dhan	21.75 ^abcd^	11.20 ^ab^	1.80 ^ab^	0.81 ^abcdef^	33.51 ^abcd^	21.83 ^abc^
Rajalakshmi	22.15 ^abcd^	12.30 ^a^	1.75 ^ab^	1.09 ^ab^	37.88 ^ab^	26.51 ^a^
Arubathamkodai	18.43 ^cde^	10.53 ^abc^	1.55 ^bc^	0.55 ^cdef^	28.42 ^cdef^	11.20 ^bcd^
Ariki raavi	19.57 ^bcde^	4.94 ^bcd^	1.31 ^d^	0.47 ^def^	25.33 ^def^	8.46 ^cd^
Chenkayama	17.53 ^de^	3.88 ^cd^	1.27 ^d^	0.33 ^f^	21.52 ^f^	5.33 ^d^
Oheruchitteni	18.26 ^cde^	4.10 ^cd^	1.33 ^cd^	0.45 ^ef^	23.77 ^ef^	5.88 ^d^
CD (0.05)	4.12	6.39	0.22	0.44	7.28	12.74

Values are mean (±standard error) (*n* = 3), and values followed by the same letter in each column are not significantly different from each other as determined by DMRT (*p <* 0.05). C—controlled; S—induced moisture stress at −1.5 MPa.

**Table 3 plants-11-00775-t003:** Mean square of ANOVA analysis for morpho-physiological and metabolic parameters in the germinated seeds.

	RWC	Proline	TSS	GA	IAA	CAT	SOD	α-Amylase	Protease	Lipase
Genotypes	1610.0 ***	8355.7 ***	5379.7 ***	44.246 ***	4211.1 ***	12.6370 ***	3908.9 ***	648.40 ***	3.8728 ***	3569.6 ***
Treatment	9005.9 ***	28,792.8***	16,172.1 ***	38.500 ***	3982.8 ***	1.0015 ***	8.4	2299.44 ***	14.9331 ***	4006.9 ***
GXT	835.1 ***	2225.1 ***	2565.0 ***	1.001 ****	1751.2 ***	6.0718 ***	54.4	158.35 ***	0.8296 ***	257.8 ***
Residuals	268.1	75.1	29.1	0.187	13.9	0.1110	274.7	7.06	0.0480	16.0

G—genotypes; T—treatment (induced moisture stress and controlled environments). *** denotes highly significant.

**Table 4 plants-11-00775-t004:** Two-way ANOVA analysis for cytokinin and ABA modulation in the germinated seeds.

Details	SS		MSS		F Value	
	Cytokinin	ABA	Cytokinin	ABA	Cytokinin	ABA
**Genotypes**	0.218	0.026	0.036	0.004	91.26 ***	20.686 ***
**Treatment**	0.076	0.041	0.079	0.041	198.46 ***	196.126 ***
**GXT**	0.009	0.006	0.001	0.001	3.85 **	5.059 **
**Residuals**	0.011	0.0066	0.001	0.002		
**CV**	6.4	12.5				
**CD(0.05)**	0.013	0.008				

Values are calculated from three replications, and * denotes level of significance. *** denotes highly significant, whereas ** is significant; G—genotypes; T—treatment (induced moisture stress and controlled environments).

## Data Availability

Data is contained within the article and Appendix A.

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
