# Peer review of "Synergistic Modulation of Seed Metabolites and Enzymatic Antioxidants Tweaks Moisture Stress Tolerance in Non-Cultivated Traditional Rice Genotypes during Germination"

_plants, 2022, doi:10.3390/plants11060775_

Round 1

Reviewer 1 Report

Dear Authors,

this work is indeed very complex and elaborate. With satisfaction I was able to read that among 100 local varieties, at least 6 can be used for future works.
Since your study is very complex, I believe that you need to revise some parts to make them clearer and more readable to the reader. I therefore suggest a revision of the English language.
Moreover,
TITLE: too long and confusing. I suspect I have a more concise title
ABSTRACT: here but also in the following sections you mention different levels of stress using PEG, but then in the text you only indicate 1. I do not understand or have not been able to grasp the motivation.
INTRODUCTION: this section explains the state of the art well. I suggest to specify better the hypotheses of the work.
M&M: the number of replicas and whether the analyzed data are homogeneous and normal and whether they have been transformed are not well specified. And how?
RESULTS: is the survival percentage the percentage of germination? Why is there no control in figure 1a?
in table 1 there are no significant differences. So it's not clear. Furthermore, the captions are generally not very descriptive. The image quality is low.

Author Response

Review Report Response

Manuscript ID: plants-1615883

Title: Synergistic modulation of seed metabolites and enzymatic antioxidants tweaks moisture stress tolerance in non-cultivated traditional rice genotypes during germination

The authors are thankful to the Editors (Prof. Chien Van Ha, Prof. Mohammad Golam Mostofa, Prof. Gopal Saha, Prof. Swarup Roy Choudhury) and Dr. Gina Wang, Section Managing Editor for devoting the time in reviewing the manuscript with constructive suggestions which become an added advantage regarding the improvement of quality of this manuscript. As a valuable suggestion, the complete manuscript has been revised and language also critically checked. Hoping for your kind consideration.

We would like to thank Respected Reviewers for taking the time and effort necessary to review the manuscript. We sincerely appreciate all valuable comments and suggestions, which helped us to improve the quality of the manuscript.

Reviewer 1:

Dear Authors,

This work is indeed very complex and elaborate. With satisfaction I was able to read that among 100 local varieties, at least 6 can be used for future works.

Since your study is very complex, I believe that you need to revise some parts to make them clearer and more readable to the reader. I therefore suggest a revision of the English language.

Reply: English revision done accordingly.

Moreover,

TITLE: too long and confusing. I suspect I have a more concise title

Reply: Title changed as “Synergistic modulation of seed metabolites and enzymatic antioxidants tweaks moisture stress tolerance in non-cultivated traditional rice genotypes during germination”.

ABSTRACT: here but also in the following sections you mention different levels of stress using PEG, but then in the text you only indicate 1. I do not understand or have not been able to grasp the motivation.

Reply: The stress levels are mentioned in the abstract. However, observations made in maximum osmotic potential of -1.5 MPa is presented in the manuscript.

INTRODUCTION: this section explains the state of the art well. I suggest to specify better the hypotheses of the work.

Reply: Hypothesis has been elaborated as suggested.

The traditional landraces that are not cultivated possess inherent potential of withstanding long spell of moisture stress, submergence tolerance, resistant to biotic stresses besides their nutraceutical properties. Germination of seeds in cultivated genotypes during early moisture stress is a major challenge whereas, landraces mitigate the moisture stress through certain physiological and metabolic modulations. We hypothesize that metabolic or physiological adaptions in non-cultivated traditional rice genotypes, accelerate their tolerance/resilience to cope up with early moisture stress. Such moisture stress-tolerant rice genotypes are a key asset for developing moisture stress-tolerant high yielding rice varieties. The study envisages the morpho-physiological and metabolic responses of traditional rice landraces from Southern Tamil Nadu during seed germination in response to induced moisture stress. 

M&M: the number of replicas and whether the analyzed data are homogeneous and normal and whether they have been transformed are not well specified. And how?

Reply: The number of replicas has been mentioned in section 2.1 under Materials and Methods as “The experiment was conducted in factorial CRD with three replications”. The data are subjected to duncan’s multiple range test to compare the mean values.

RESULTS: is the survival percentage the percentage of germination? Why is there no control in figure 1a?

In table 1 there are no significant differences. So it's not clear. Furthermore, the captions are generally not very descriptive. The image quality is low.

Reply: The survival percent is the percentage of germination observed at induced stress using PEG. Control is non-stressed environment. In Fig 1a, both stress and control are depicted clearly. In table 1, significant differences observed in vigor index, RWC, shoot length, RS ratio which are major criterion that differed significantly among genotypes. Moreover, IR64 and IR 64 (Drt 1) are used drought susceptible and drought tolerant checks.

Reviewer 2 Report

Authors evaluated 100 rice genotypes including 85 traditional landraces from Southern Tamil Nadu and 15 improved cultivars in relation to future development of climate-resilient rice cultivars. Authors analyzed moisture stress-susceptible and moisture stress-tolerant genotypes, and particularly landraces were screened over a range of osmotic potentials. Authors assessed rate of germination, root and shoot length, vigor index, R/S ratio, relative water content, seed macromolecules, phytohormones, osmolytes, and enzymatic antioxidants. Authors hypothesize that metabolic or physiological adaptions in non-cultivated traditional rice genotypes, accelerate their tolerance/resilience to cope up with early moisture stress.

Major points:

  1. Define “seed macromolecules” did you mean storage materials?, please precise it.
  2. Use precisely terms “germinated seeds” and “during seed germination” Both were used in the text. Germination sensu stricto ends when a radicle is emerged, therefore the second phrase is rather non proper here. Consider this suggestion because none analyses were conducted between imbibition and radicle protrusion stage.
  3. 71-73 add a reference and a species to which this composition of chemicals is given because seeds greatly differ in storage material composition.
  4. 190 and 197, precise whether one specific enzyme was analyzed or a group, then change headings to plural.
  5. 204 an opening bracket is missing or the second closing one is unnecessary, please check the formula.
  6. Figure 1 – “a)” and “b)” are given twice; l.283 “percent” or “percentage”; line 284 repeats line 283 (it repeats in each figure caption, is it necessary?)
  7. Table 1 – add statistics.
  8. Which figure/table shows proline content? Table 3 p. 14 includes statistics but the results appears on page 19, please sort the results.
  9. Figure 6 – present the columns as cat-c, cat-s, sod-c, sod-s, and not as cat-c, sod-c, sod-s, cat-s, as it is estimated from the legend; Figure 4 and 7 as well should be redone.
  10. Figure 5 – chromatograms are blurry and should be transferred in better resolution to supplemental data. Retention time is not visible. Single runs are not informative. Means from biological repetitions should be only presented.
  11. Figure 9, exclude the diagonal where symbols means R=1 between the same parameters, increase the font of parameters and resize the matrices because the symbols are not round but squeezed, replace hyphen with minus on scale

Minor points:

Abstract contains many unexplained abbreviations (GA, IAA, ABA, CAT, SOD, HPLC, ROS)

Reference numbers are given in wrong format, use’[]” instead of  “()”

l.41 catalase is used after abbreviation in line 39

l.73 a space is needed “stresses(drought”

l.80 explain the use of "Kreb’s cycle" instead of “Krebs cycle”

l.87 rewrite the beginning of this sentence “Cysteine proteases are located (15), participating”

l.106, provide a full name before abbreviation of H2O2

l.133 the correct abbreviation for minute(s) is "min", it does not change in plural

l.137 use a minus sign not hyphen in the whole text

l.138, explain abbreviation CRD

l.139 change “28oC” into “28 °C”; l.168 also 195 and in the whole text

l.150 “Relative Water Content” or “relative water content” or “RWC”

l.154 a comma is not necessary

l.161 format “–1” to superscript

l.162 a comma is not necessary, also l.185, 191 and further in the whole text

l.169 and 170 a hyphen and minus are in the superscript, similarly l.180 and 181

l.173 “was grounded”

l.192 a space is needed “mMTris”

l.194, 212 did you mean “was added” or ‘we added”, please rephrase

l.208 previously “germinated seeds” stage was described as used in methods sections

l.216 full name is used after introducing an abbreviation

l.221, 223, 243 and others  “1mL” previously a space was used between the number and unit”, unify it in the whole text

l.223, 234 µl or µL?, unify the format in the whole text

l.229 delete the first “at”

l.230 authors previously introduced CKs and use full name in the nieghbourhood of abbreviations of other enzymes

l.242 a space is needed “5DAI”

l.244 centrifugation is expresses as “xg” and previously in rpm, unity them

l.277  a space is needed “ ).The”

l.291-294, remove numbers from text which are visible in the table, similarly l. 300-302 and further in the description of results

figure 2c Y-axis name:  “amino acids”

l.338, “genotypes.( Influence” correct the place of space

l.369 "lipases.NS" Add a space

l.400 “CKs” was already  introduced

l.407 superscript format is needed

l.415 please correct “S7. .a)”

l.418 abbreviation was already introduced, please check whole the text for all abbreviations used

l.477, 486  what does “viz.” mean? it appears 5 times in the manuscript

l.479 “Phytohormones” or “phytohormones”

l.480 “control condition (C)” this abbreviations was introduced and used before

l.490 “antioxidant gadgets” in not a good phrase

l.589 “was noticed”?

l.610 use only abbreviations

l.631 use abbreviation

l.633 abbreviation was not explained before

l.706 provide authors contribution categories provided by the MDPI system

l. 710, Authors wrote that none external funding was applied but gave two grant numbers in Acknowledgements, please explain

l.782 italicize species latin name, check other references

Author Response

Review Report Response

Manuscript ID: plants-1615883

Title: Synergistic modulation of seed metabolites and enzymatic antioxidants tweaks moisture stress tolerance in non-cultivated traditional rice genotypes during germination

The authors are thankful to the Editors (Prof. Chien Van Ha, Prof. Mohammad Golam Mostofa, Prof. Gopal Saha, Prof. Swarup Roy Choudhury) and Dr. Gina Wang, Section Managing Editor for devoting the time in reviewing the manuscript with constructive suggestions which become an added advantage regarding the improvement of quality of this manuscript. As a valuable suggestion, the complete manuscript has been revised and language also critically checked. Hoping for your kind consideration.

We would like to thank Respected Reviewers for taking the time and effort necessary to review the manuscript. We sincerely appreciate all valuable comments and suggestions, which helped us to improve the quality of the manuscript.

Reviewer 2:

Authors evaluated 100 rice genotypes including 85 traditional landraces from Southern Tamil Nadu and 15 improved cultivars in relation to future development of climate-resilient rice cultivars. Authors analyzed moisture stress-susceptible and moisture stress-tolerant genotypes, and particularly landraces were screened over a range of osmotic potentials. Authors assessed rate of germination, root and shoot length, vigor index, R/S ratio, relative water content, seed macromolecules, phytohormones, osmolytes, and enzymatic antioxidants. Authors hypothesize that metabolic or physiological adaptions in non-cultivated traditional rice genotypes, accelerate their tolerance/resilience to cope up with early moisture stress.

Major points:

Define “seed macromolecules” did you mean storage materials?, please precise it.

Reply: Yes. Seed macromolecules refers to the biomolecules like starch, protein and lipids. The hydrolytic enzymes and their activities in converting these complex molecules  into their respective simple molecules were studied. The bioconversion of seed macromolecules is a typical biochemical changes happening in seeds during germination process.

Use precisely terms “germinated seeds” and “during seed germination” Both were used in the text. Germination sensu stricto ends when a radicle is emerged, therefore the second phrase is rather non proper here. Consider this suggestion because none analyses were conducted between imbibition and radicle protrusion stage.

Reply: The suggestion has been considered and changed accordingly in the manuscript. 

71-73 add a reference and a species to which this composition of chemicals is given because seeds greatly differ in storage material composition.

Reply: Reference added

190 and 197, precise whether one specific enzyme was analyzed or a group, then change headings to plural.

Reply: The headings changed plural.

204 an opening bracket is missing or the second closing one is unnecessary, please check the formula.

Figure 1 – “a)” and “b)” are given twice; l.283 “percent” or “percentage”; line 284 repeats line 283 (it repeats in each figure caption, is it necessary?)

Reply: Formula checked and changed. Corrections made

Table 1 – add statistics.

Reply: The mean values are compared already using DMRT since it involves 100 genotypes. However, statistics (ANOVA) is included in further analysis.

Which figure/table shows proline content? Table 3 p. 14 includes statistics but the results appears on page 19, please sort the results.

Reply: To avoid more tables, the ANOVA of proline was given in Table 3 along with other parameters.  The results are given under subsection osmolytes.

Figure 6 – present the columns as cat-c, cat-s, sod-c, sod-s, and not as cat-c, sod-c, sod-s, cat-s, as it is estimated from the legend; Figure 4 and 7 as well should be redone.

Reply: The figure in the present form gives more information on the performance of the same genotype under stressed and non-stressed environment. Here we are comparing the genotypes with the environment. Hence it is not necessary to redo the figures.

Figure 5 – chromatograms are blurry and should be transferred in better resolution to supplemental data. Retention time is not visible. Single runs are not informative. Means from biological repetitions should be only presented.

Reply: Chromatograms shifted to supplemental data. The data given are mean values of three runs. The chromatograms presented represents the respective standards. The retention times differs with phytohormones. The RT for ABA, trans zeatin, and BAP is 2 min, IAA – 1.7 min, Kinetin and zeatin is 11.63 min.

Figure 9, exclude the diagonal where symbols means R=1 between the same parameters, increase the font of parameters and resize the matrices because the symbols are not round but squeezed, replace hyphen with minus on scale

Reply: Fig 9 is the correlation plot obtained using R software. The symbol (-) in the scale is not hyphen. It shows minus scale.

Minor points:

Abstract contains many unexplained abbreviations (GA, IAA, ABA, CAT, SOD, HPLC, ROS)

Reply: Changes made as suggested

Reference numbers are given in wrong format, use’[]” instead of  “()”

Reply: Changed

l.41 catalase is used after abbreviation in line 39

Reply: changed

l.73 a space is needed “stresses(drought”

Reply: Changed

l.80 explain the use of "Kreb’s cycle" instead of “Krebs cycle”

Reply: Use of Krebs cycle added

l.87 rewrite the beginning of this sentence “Cysteine proteases are located (15), participating”

Reply: Rephrased

l.106, provide a full name before abbreviation of H2O2

Reply: Revised

l.133 the correct abbreviation for minute(s) is "min", it does not change in plural

Reply: Revised

l.137 use a minus sign not hyphen in the whole text

Reply: minus sign used

l.138, explain abbreviation CRD

Reply: Revised

l.139 change “28oC” into “28 °C”; l.168 also 195 and in the whole text

Reply: Changed

l.150 “Relative Water Content” or “relative water content” or “RWC”

Reply: relative water content (RWC)

l.154 a comma is not necessary

Reply: Changed

l.161 format “–1” to superscript

Reply: changed

l.162 a comma is not necessary, also l.185, 191 and further in the whole text

Reply: Changed

l.169 and 170 a hyphen and minus are in the superscript, similarly l.180 and 181

Reply: changed

l.173 “was grounded”

Reply: corrected

l.192 a space is needed “mMTris”

Reply: corrected

l.194, 212 did you mean “was added” or ‘we added”, please rephrase

Reply: corrected

l.208 previously “germinated seeds” stage was described as used in methods sections

Reply: corrected

l.216 full name is used after introducing an abbreviation

Reply: corrected

l.221, 223, 243 and others  “1mL” previously a space was used between the number and unit”, unify it in the whole text

Reply: corrected

l.223, 234 µl or µL?, unify the format in the whole text

Reply: corrected

l.229 delete the first “at”

Reply: corrected

l.230 authors previously introduced CKs and use full name in the nieghbourhood of abbreviations of other enzymes

Reply: corrected

l.242 a space is needed “5DAI”

Reply: corrected

l.244 centrifugation is expresses as “xg” and previously in rpm, unity them

Reply: corrected

l.277  a space is needed “ ).The”

Reply: Checked

l.291-294, remove numbers from text which are visible in the table, similarly l. 300-302 and further in the description of results

Reply: corrected

figure 2c Y-axis name:  “amino acids”

Reply: corrected

l.338, “genotypes.( Influence” correct the place of space

Reply: corrected

l.369 "lipases.NS" Add a space

Reply:corrected

l.400 “CKs” was already  introduced

Reply: corrected

l.407 superscript format is needed

Reply: corrected

l.415 please correct “S7. .a)”

Reply: corrected

l.418 abbreviation was already introduced, please check whole the text for all abbreviations used

Reply: corrected

l.477, 486  what does “viz.” mean? it appears 5 times in the manuscript

Reply: corrected

l.479 “Phytohormones” or “phytohormones”

Reply: corrected

l.480 “control condition (C)” this abbreviations was introduced and used before

Reply: corrected

l.490 “antioxidant gadgets” in not a good phrase

Reply: changed as antioxidant enzymes

l.589 “was noticed”?

Reply: corrected

l.610 use only abbreviations

Reply: corrected

l.631 use abbreviation

Reply: corrected

l.633 abbreviation was not explained before

Reply: corrected

l.706 provide authors contribution categories provided by the MDPI system

Reply: Thank you for your query, we have already submitted the Authorship Contribution Form to editorial office.

  1. 710, Authors wrote that none external funding was applied but gave two grant numbers in Acknowledgements, please explain

Reply: Thank you for your query and we have revised the information. 

l.782 italicize species latin name, check other references

Reply: corrected

Round 2

Reviewer 1 Report

Dear Authors,

it seems to me that every request from the auditors has been adequately paid.
Congratulations on your work.

Author Response

Manuscript ID: plants-1615883

Title: Synergistic modulation of seed metabolites and enzymatic antioxidants tweaks moisture stress tolerance in non-cultivated traditional rice genotypes during germination

Reviewer 1:

Dear Authors,

It seems to me that every request from the auditors has been adequately paid. Congratulations on your work.

Reply: Thank you so much for your recommendation. We really appreciate it.

Reviewer 2 Report

1. Please consider the difference between a hyphen and minus sign.
Authors used minus in line 833, in the majority of other places o a hyphen is used instead of minus.

2. Authors did not check the text carefully. Text still contains many editorial mistakes.

l.194
"28oC" change to "28 °C"

l.323 superoxide dismutase
l.471 "50.27μ.mol g-1FW" ad spaces where needed
l.507 "-1" should be in superscript (all superscripts also must contain minus not hyphen)

l.570 "μg/g" and "μg.g-1" in the same line
l.833 'MPa" and a space before
Figure 1
"a)" is given twice, "b)" also

Figure 2, the name of Y axis is still "Aminoacids" instead of "Amino acids"

Figure 9
you can use "diag=FALSE" in R 
R is statistical program, therefore the change of names of analyzed parameters must be done in a graphic program.
Improve the names: 
- R is using "." instead of a space,
- increase size and/or use bold format.
The scale is from minus one to plus one, but..... still R uses a hyphen to show the minus values.
Still the circle symbols are squeezed. R produces matrix with simmetrical circles.

l.1038 italicize Latin name
1045 "2" should be changed to subscript format
l.934, l.1048 full journal name is given without italicized format

Author Response

Reviewer 2:

Thank you for your suggestion and correction. We really appreciate it. As per your suggestion, the whole manuscript has been revised. We firmly believe that immensely improved our manuscript after incorporation of your most valuable suggestion and comments. Hoping for your kind consideration.

  1. Please consider the difference between a hyphen and minus sign.
    Authors used minus in line 833, in the majority of other places o a hyphen is used instead of minus.

Reply: Thank you and checked

  1. Authors did not check the text carefully. Text still contains many editorial mistakes.

l.194 "28oC" change to "28 °C"

Reply: Thank you and revised.

l.323 superoxide dismutase

Reply: Revised.

l.471 "50.27μ.mol g-1FW" ad spaces where needed

Reply: Checked

l.507 "-1" should be in superscript (all superscripts also must contain minus not hyphen)

Reply: Checked

l.570 "μg/g" and "μg.g-1" in the same line
Reply: Revised

l.833 'MPa" and a space before

Reply: Checked

Figure 1
"a)" is given twice, "b)" also

Reply: Thank you and revised.

Figure 2, the name of Y axis is still "Aminoacids" instead of "Amino acids"

Reply: Revised

Figure 9
you can use "diag=FALSE" in R 
R is statistical program, therefore the change of names of analyzed parameters must be done in a graphic program.
Improve the names: 
- R is using "." instead of a space,
- increase size and/or use bold format.
The scale is from minus one to plus one, but..... still R uses a hyphen to show the minus values.
Still the circle symbols are squeezed. R produces matrix with simmetrical circles.

Reply: Thank you for your valuable suggestion and as per your suggestions we have revised the figures.

l.1038 italicize Latin name
Reply: Revised

1045 "2" should be changed to subscript format
Reply: Revised

l.934, l.1048 full journal name is given without italicized format

Reply: Thank you and revised
